# An alternative splicing switch in FLNB promotes the mesenchymal cell state in human breast cancer

Ji Li[1,2,3†], Peter S Choi[1,2,3†], Christine L Chaffer[4,5], Katherine Labella[1,2], Justin H Hwang[1,2,3], Andrew O Giacomelli[1,2,3], Jong Wook Kim[1,2,3], Nina Ilic[1,2,3], John G Doench[3], Seav Huong Ly[1,2,3], Chao Dai[1,2,3], Kimberly Hagel[1,2], Andrew L Hong[1,2,3], Ole Gjoerup[1,2,3], Shom Goel[2,6], Jennifer Y Ge[1,2,7], David E Root[3], Jean J Zhao[2,6], Angela N Brooks[8], Robert A Weinberg[4], William C Hahn[1,2,3]*

[1]Department of Medical Oncology, Dana-Farber Cancer Institute, Boston, United States; [2]Harvard Medical School, Boston, United States; [3]Broad Institute of MIT and Harvard, Cambridge, United States; [4]Whitehead Institute for Biomedical Research and MIT, Cambridge, United States; [5]Garvan Institute of Medical Research, Sydney, Australia; [6]Department of Cancer Biology, Dana-Farber Cancer Institute, Boston, United States; [7]Department of Biostatistics and Computational Biology, Dana-Farber Cancer Institute, Boston, United States; [8]University of California, Santa Cruz, Santa Cruz, United States

*For correspondence:
william_hahn@dfci.harvard.edu

†These authors contributed equally to this work

Competing interests: The authors declare that no competing interests exist.

**Abstract** Alternative splicing of mRNA precursors represents a key gene expression regulatory step and permits the generation of distinct protein products with diverse functions. In a genome-scale expression screen for inducers of the epithelial-to-mesenchymal transition (EMT), we found a striking enrichment of RNA-binding proteins. We validated that QKI and RBFOX1 were necessary and sufficient to induce an intermediate mesenchymal cell state and increased tumorigenicity. Using RNA-seq and eCLIP analysis, we found that QKI and RBFOX1 coordinately regulated the splicing and function of the actin-binding protein FLNB, which plays a causal role in the regulation of EMT. Specifically, the skipping of FLNB exon 30 induced EMT by releasing the FOXC1 transcription factor. Moreover, skipping of FLNB exon 30 is strongly associated with EMT gene signatures in basal-like breast cancer patient samples. These observations identify a specific dysregulation of splicing, which regulates tumor cell plasticity and is frequently observed in human cancer.

DOI: https://doi.org/10.7554/eLife.37184.001

## Introduction

Alternative splicing (AS) of mRNA precursors is a fundamental biological process that provides a reversible mechanism to modulate the expression of related but distinct proteins in response to internal and external stimuli (*Chen and Manley, 2009*). Regulation of alternative splicing occurs at several levels including the expression and the targeting of specific RNA-binding proteins (RBPs). Dysregulation of alternative splicing plays a direct role in a variety of human diseases including cancer (*David and Manley, 2010*).

During cancer initiation and progression, the epithelial-to-mesenchymal transition (EMT) triggers the dissociation and migration of carcinoma cells from primary to distant sites (*Ye and Weinberg, 2015*). We previously demonstrated that the EMT is also tightly linked to a stem-like cell state in breast cancer, as overexpression of EMT transcription factors induces the expression of tumor-

**eLife digest** As the human body develops, countless cells change from one state into another. Two important cell states are known as epithelial and mesenchymal. Cells in the epithelial state tend to be tightly connected and form barriers, like skin cells. Mesenchymal state cells are loosely organized, move around more and make up connective tissues. Some cells alternate between these states via an epithelial-to-mesenchymal transition (EMT for short) and back again. Without this transition, certain organs would not develop and wounds would not heal. Yet, cancer cells also use this transition to spread to distant sites of the body. Such cancers are often the most aggressive, and therefore the most deadly.

The epithelial-to-mesenchymal transition is dynamically regulated in a reversible manner. For example, the genes for some proteins might only be active in the epithelial state and further reinforce this state by turning on other 'epithelial genes'. Alternatively, there might be differences in the processing of mRNA molecules – the intermediate molecules between DNA and protein – that result in the production of different proteins in epithelial and mesenchymal cells.

Li, Choi et al. wanted to know which of the thousands of human genes can endow epithelial state cells with mesenchymal characteristics. A better understanding of the switch could help to prevent cancers undergoing an epithelial-to-mesenchymal transition.

From a large-scale experiment in human breast cancer cells, Li, Choi et al. found that a group of proteins that bind and modify mRNA molecules are important for the epithelial-to-mesenchymal transition. Two proteins in particular promoted the transition, most likely by binding to the mRNA of a third protein called FLNB and removing a small piece of it. FLNB normally works to prevent the epithelial-to-mesenchymal transition, but the smaller protein encoded by the shorter mRNA promoted the transition by turning on 'mesenchymal genes'.

This switching between different FLNB proteins happens in some of the more aggressive breast cancers, which also contain mesenchymal cells. Finding out which FLNB protein is made in a given cancer may provide an indication of its aggressiveness. Also, looking for drugs that can target the mRNA-binding proteins or FLNB may one day lead to new treatments for some of the most aggressive breast cancers.

DOI: https://doi.org/10.7554/eLife.37184.002

initiating cell markers and increases the ability of cells to form mammospheres, a property associated with mammary epithelial stem cells (*Chaffer et al., 2013*; *Mani et al., 2008*). In addition, the EMT has been implicated in several other cancer-related phenotypes, for example, in cancers that acquired resistance either to the EGFR inhibitor gefitinib or to the HER2 receptor inhibitor trastuzumab (*Boulbes et al., 2015*; *Sequist et al., 2011*).

EMT also involves a dramatic reorganization of the actin cytoskeleton and concomitant formation of membrane protrusions to gain migratory and invasive properties (*Yilmaz and Christofori, 2009*). The dynamic change in the actin cytoskeleton, a prerequisite for cell motility and cancer cell invasion, is a highly controlled equilibrium of local assembly and disassembly of actin filaments (*Yilmaz and Christofori, 2009*). The filamin family proteins crosslink actin filaments and are also translocated into the nucleus to regulate the transcriptional activity of the androgen receptor and the FOXC1 transcription factor (*Bedolla et al., 2009*; *Berry et al., 2005*; *Loy et al., 2003*; *Zhou et al., 2010*). The three members of this family (FLNA, FLNB and FLNC) are involved in both development and normal tissue homeostasis through regulating diverse processes including cell locomotion and integrin signaling (*Zhou et al., 2010*), and mutations in the *FLNB* gene cause a broad range of skeletal dysplasias (*Daniel et al., 2012*).

Alternative splicing has been previously associated with EMT. Mesenchymal cancer cells show distinct alternative splicing patterns in comparison with their epithelial counterparts (*Braeutigam et al., 2014*; *Shapiro et al., 2011*; *Venables et al., 2013*). While ESRP1 and ESRP2 are epithelial state-inducing RBPs that govern splicing patterns for the epithelial cell state (*Shapiro et al., 2011*; *Warzecha et al., 2010*; *Warzecha et al., 2009*; *Yang et al., 2016*), less is known about the identity and functional significance of RBPs that can promote the mesenchymal cell state. QKI and RBFOX2 have been shown to be responsible for alternative splicing events that occur during EMT, such as

exon skipping in KIF13A and CTTN (*Braeutigam et al., 2014*; *Venables et al., 2013*; *Yang et al., 2016*) and in circular RNA formation (*Conn et al., 2015*). Nevertheless, it remains unclear whether the upregulation of any specific RBPs is sufficient or required for the induction of mesenchymal state transitions or is merely one of many downstream manifestations of the EMT. Furthermore, although many splicing changes occur during EMT, only a small number of specific splicing events are known to functionally contribute to EMT including changes in the splicing of CD44, FGFR2 and Exo70 (*Brown et al., 2011*; *Lu et al., 2013*; *Warzecha et al., 2009*). Here, we have undertaken a comprehensive approach to identify genes that regulate the EMT in breast cancer and found that genes whose protein products participate in AS regulate the transition to mesenchymal- and stem-like cell states.

## Results

### A genome scale ORF screen to identify regulators of the mesenchymal cell state

In prior work, we described a genetically defined, experimental model of breast cancer, derived from introducing vectors expressing the telomerase catalytic subunit, the SV40 large-T and small-t antigens, and an H-Ras oncoprotein into human mammary epithelial cells (HMLER cells) (*Elenbaas et al., 2001*). Subsequent work demonstrated that the CD44 cell surface antigen is a surrogate marker for the EMT cell state change in this model (*Chaffer et al., 2011*; *Chaffer et al., 2013*). Thus, we separated the CD44-high and -low populations of HMLER cells by fluorescence-activated cell sorting (FACS) and confirmed that the CD44-low cells displayed epithelial properties, as measured by levels of EMT marker expression (*Figure 1—figure supplement 1A*). The highly purified CD44-low cell population remained in the epithelial cell state for at least 4 weeks in the experimental conditions. In contrast, the CD44-high HMLER cells showed elevated expression of mesenchymal markers and a greater propensity to form mammospheres, an in vitro surrogate assay for the stemness of mammary epithelial cells (*Figure 1—figure supplement 1B,C*).

To study inducers of the EMT and stem-like cell state, we performed a genome scale open-reading frame (ORF) screen to identify genes that convert the HMLER cells from the CD44-low state to the CD44-high state. Each ORF in the human ORFeome library collection 8.1 (*Yang et al., 2011*) was tagged with a unique 24-nucleotide barcode and introduced into FACS purified CD44-low HMLER cells by lentiviral-mediated gene transfer. Following 7 days in culture, we purified the newly arising CD44-high HMLER cells by FACS and identified ORFs enriched in these cells by massively parallel sequencing (*Figure 1A*).

We found that the consistency between the biological replicates of the screen was high (*Figure 1—figure supplement 2A*). *SNAI1*, a well-characterized EMT-inducing transcription factor (EMT-TF) (*Nieto et al., 2016*), scored as the top hit in the screen, as did *BCL6*, *JMJD6* and *FOS*, which have previously been shown to play key roles in regulating EMT (*Aprelikova et al., 2016*; *Eger et al., 2000*; *Yu et al., 2015*) (*Figure 1B*, *Figure 1—figure supplement 2B* and *Figure 1—source data 1*); these findings indicated that this screen was robust. We used a cut-off of three standard deviations (S.D.) above the mean and analyzed the top-scoring candidates to identify protein complexes or pathways enriched for regulators of EMT. Sixty-eight ORFs met this criterion (*Figure 1B* and *Figure 1—figure supplement 2B*), including transcription factors, RNA splicing factors, kinases and phosphatases, epigenetic regulators, and genes involved in the regulation of spermatogenesis, apoptosis and the metabolic processing of cellular amides (*Figure 1C*). Other EMT transcription factors did not meet the 3 S.D. cutoff possibly due to mutations in the ORF constructs or low ORF representation in the library.

Using the GeNets analysis tool (*apps.broadinstitute.org/genets*), we found three gene networks centered around *QKI* (Quaking, an RNA-binding protein), *SRPK2* (a kinase involved in RNA splicing) and *PPP1CC* (a phosphatase) (*Figure 1D*). When we used the top candidates to interrogate gene ontology, we found that the 'regulation of mRNA metabolic process' and 'regulation of mRNA splicing, via spliceosome' scored as the top terms (*Figure 1E*) and that 'RNA metabolic process' was one of the top gene sets enriched by gene set enrichment analysis (GSEA) (*Figure 1F*) (*Reich et al., 2006*). Of note, 'Regulation of mRNA metabolic process' is a parent GO term for RNA processing and RNA splicing.

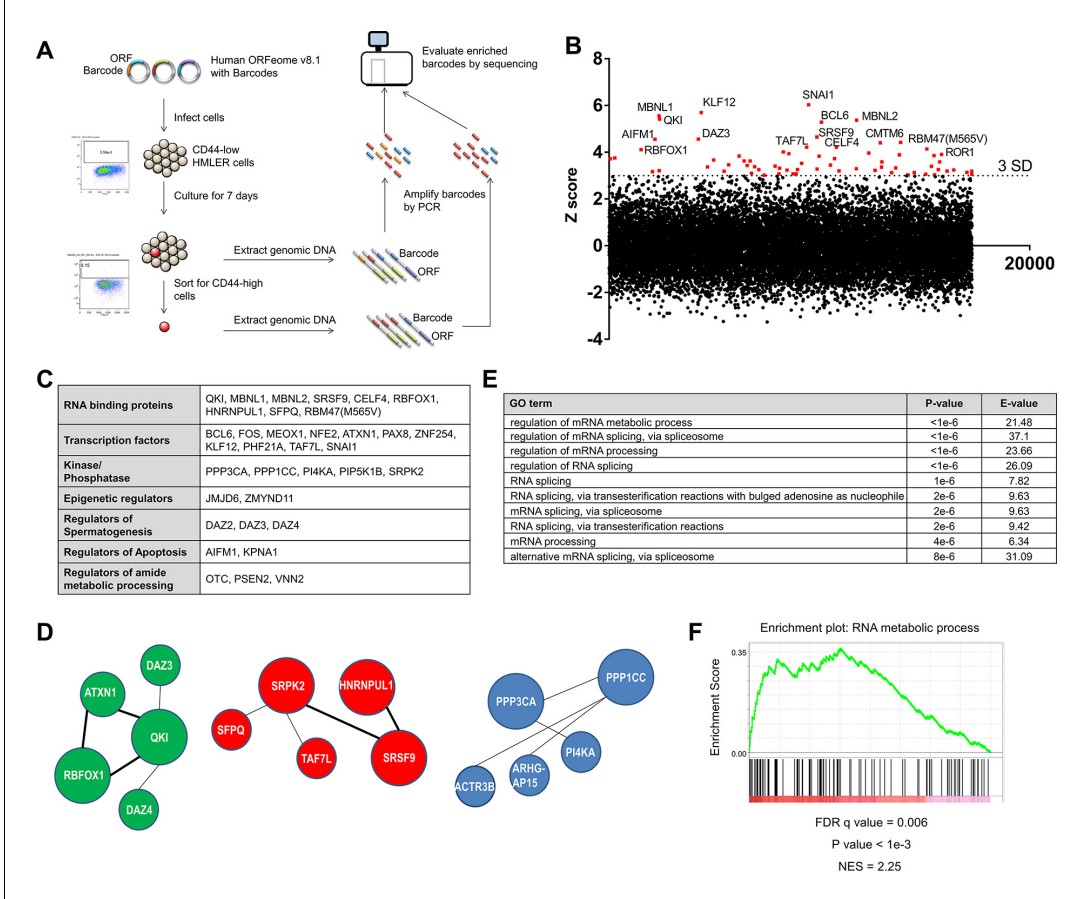

**Figure 1.** Genome scale ORF screen identifies splicing factors and RNA-binding proteins as regulators of EMT. (**A**) Schematic of the genome scale ORF screen used to identify regulators of EMT. (**B**) Distribution of enrichment Z scores. The red dots indicate the ORFs with a Z score >3 and the top 15 gene names are labeled (***Figure 1—source data 1***). (**C**) The top candidate ORFs in the screen categorized into seven different functional classes. (**D**) Network analysis of enriched protein complexes among the top candidates ORFs. (**E**) GO term enrichment analysis of the top candidate ORFs. (**F**) Enrichment for RNA metabolic processes among the top candidate ORFs, as determined by pre-ranked GSEA analysis.

DOI: https://doi.org/10.7554/eLife.37184.003

The following source data and figure supplements are available for figure 1:

**Source data 1.** ORF screen results.

DOI: https://doi.org/10.7554/eLife.37184.006

**Figure supplement 1.** CD44 status serves as a marker for the mesenchymal and stem-like cell state in human mammary epithelial cells.

DOI: https://doi.org/10.7554/eLife.37184.004

**Figure supplement 2.** A genome scale ORF screen to identify regulators of EMT.

DOI: https://doi.org/10.7554/eLife.37184.005

Several RNA-binding proteins have been previously associated with EMT. For example, ESRP1 and 2 promote an epithelial phenotype, while QKI and RBFOX2 (a homolog of RBFOX1 that scored in the screen) regulate a number of EMT-associated splicing events (***Braeutigam et al., 2014***; ***Venables et al., 2013***; ***Yang et al., 2016***). Of note, although RBFOX2 has been shown to play a role in EMT (***Braeutigam et al., 2014***; ***Venables et al., 2013***), the RBFOX2 clone present in the ORFeome collection 8.1 library harbors three mutations (a 396–449 deletion, a 752–763 deletion and a C to T substitution at 1007), which likely explained why this ORF did not score in the screen. However, whether the expression of any RBPs is functionally sufficient or required to induce a mesenchymal cell state remains unclear. Since we found a striking enrichment of RBPs in this screen, we focused on the top candidates implicated in pre-mRNA splicing to understand their possible role in regulating the EMT and stem-like cell states in breast cancer pathogenesis.

## Expression of QKI and RBFOX1 are necessary and sufficient to induce an intermediate mesenchymal cell state

In the ORF expression screen, we identified eight candidate RBPs (QKI, RBFOX1, MBNL1, MBNL2, CELF4, SFPQ, SRSF9 and HNRNPUL1) that scored when tested individually. We systematically tested these genes in five assays to examine EMT-associated phenotypes or marker expression to find the RBPs that meet the following criteria: (1) Expression of the RBP promotes an increase in the CD44-high population (*Figure 2—figure supplement 1A*); (2) Expression of the RBP upregulates the expression of a panel of mesenchymal markers examined by both quantitative PCR (*Figure 2A* and *Figure 2—figure supplement 1B*) and immunoblotting (*Figure 2B*); (3) Expression of the RBP induces mammosphere formation when cells are grown in suspension, a characteristic of the stem-like and mesenchymal cell properties (*Figure 2C* and *Figure 2—figure supplement 1C*) (*Chaffer et al., 2013*; *Mani et al., 2008*); (4) Endogenous expression of the RBP is upregulated upon overexpression of an EMT-inducing transcription factor, SNAI1 or ZEB1 (*Figure 2—figure supplement 2A–D*); (5) Expression of the RBP promotes tumor formation in vivo, a feature associated with stem-like cells (*Figure 2D* and*Figure 2—figure supplement 2*; *Figure 2—figure supplement 2E*) (*Chaffer et al., 2013*; *Mani et al., 2008*). Together, we discovered that the expression of QKI (NCBI Reference: NM_006775.2, also known as QKI-5) and RBFOX1 (NCBI Reference: NM_145893.2, also known as RBFOX1 beta) strongly induced the mesenchymal and stem-like phenotypes in all the experiments tested, while MBNL1, MBNL2 and CELF4 scored in some assays. We also found that SRSF9, SFPQ and HNRNPUL1 are unlikely to initiate a mesenchymal and stem-like cell state (*Figure 2J*). The CD44-high cells induced by QKI, RBFOX1 or SNAI1 shared a similar elongated and spindle shape cell morphology (*Figure 2—figure supplement 3A*). In addition, QKI and RBFOX1 overexpression also significantly increased the CD44-high cell populations in two additional breast cancer cell lines (MCF7 and ZR75-1) (*Figure 2—figure supplement 3B*). We thus proceeded to focus on the role of QKI and RBFOX1 in EMT.

Of note, overexpression of QKI and RBFOX1 reduced cell proliferation by 40% to 45% as would be expected if the cells undergo an EMT (*Figure 2—figure supplement 3C*) (*Tsai et al., 2012*; *Vega et al., 2004*). In addition, the expression of QKI, RBFOX1 and other RBPs failed to decrease the expression of epithelial markers (*Figure 2A,B* and *Figure 2—figure supplement 1B*), suggesting that the cell state triggered by expression of the RBPs involves elevated expression of mesenchymal markers with retention of pre-existing epithelial marker expression. This spectrum of marker expression is reminiscent of an intermediate EMT state that is implicated in development and tumor progression (*Bierie et al., 2017*; *George et al., 2017*; *Nieto et al., 2016*; *Schmidt et al., 2015*).

To determine whether expression of QKI or RBFOX1 was also required for the induction of an EMT program, we silenced endogenous QKI or RBFOX1 by short hairpin RNA (shRNA)-mediated suppression and by CRISPR/Cas9-mediated knockout. First, we expressed the SNAI1 EMT-TF to induce EMT and then depleted QKI, RBFOX1 or other candidate RBPs with shRNAs (shSNAI1 as a positive control) (*Figure 2E* and *Figure 2—figure supplement 3D*). shRNA-mediated suppression of QKI and RBFOX1 led to a significant reduction in the CD44-high cell population (*Figure 2E*), suggesting that the expression of QKI and RBFOX1 was partially required for the induction of the CD44-high cell population after SNAI1 overexpression. To eliminate the potential off-target effects of the shRNAs, we used CRISPR/Cas9 to target the QKI and RBFOX1 genes and found that the ablation of QKI and RBFOX1 also significantly suppressed the induction of CD44-high cells (*Figure 2F*), the expression of mesenchymal markers (*Figure 2G,H*) and the formation of mammospheres (*Figure 2I*) after SNAI1 overexpression. Thus, these loss-of-function studies revealed that QKI and RBFOX1 are partially required for induction of the EMT.

We next examined whether the expression of QKI and RBFOX1 also correlated with mesenchymal features in murine or human tumor samples. We discovered that the expression of QKI was highly upregulated in mesenchymal breast tumor patient samples available from the Cancer Genome Atlas (TCGA) (*Ciriello et al., 2015*) (*Charafe-Jauffret et al., 2006*) (*Figure 2—figure supplement 4A* and the Materials and methods section for data analysis). The lack of a significant change in expression of RBFOX1 suggested that QKI instead may play a major role in driving the alternative splicing patterns in these samples. In addition, both Qk and Rbfox1 are highly associated with the activation of an EMT program in a murine mammary tumor model (*Figure 2—figure supplement 4B–D* and the Materials and methods section) (*Goel et al., 2016*).

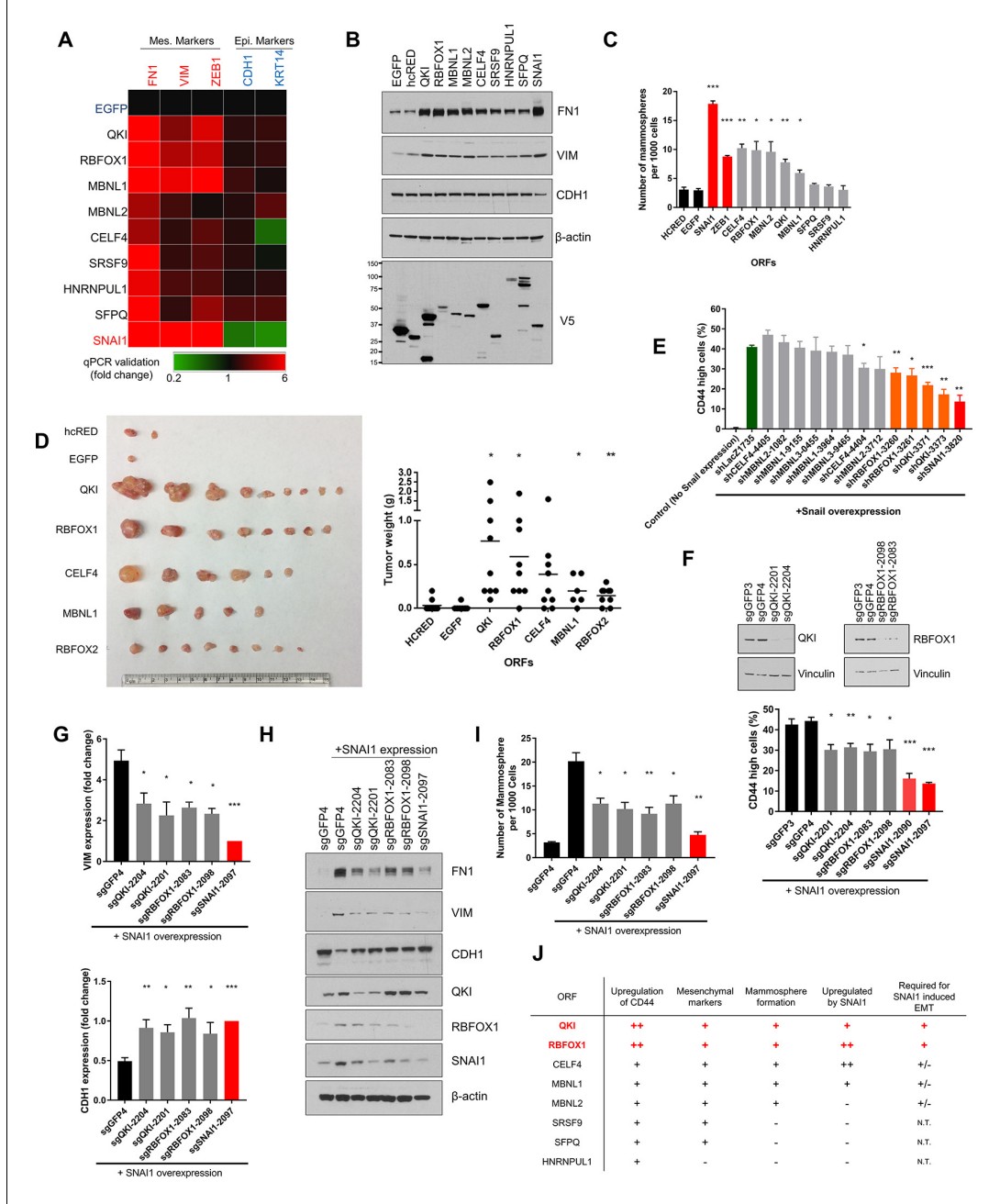

**Figure 2.** QKI and RBFOX1 are sufficient and partially required for an intermediate mesenchymal and stem-like cell state. (**A**) Heatmap showing the expression of mesenchymal (FN1, VIM, ZEB1) and epithelial (CDH1, KRT14) markers in HME cells expressing the indicated candidate ORFs, normalized to a negative control (EGFP) and quantified by qPCR. n = 3. Bar graphs and statistical significance for each gene are shown in *Figure 2—figure supplement 1B*. (**B**) Levels of EMT marker expression in HME cells expressing the indicated candidate ORFs quantified by immunoblotting. (**C**) The rate of mammosphere formation in HMLE cells that express the indicated ORFs. The number of mammosphere per 1000 cells plated was counted. n = 5, ***, p<0.001; **, p<0.01; *, p<0.05. Representative images are shown in *Figure 2—figure supplement 1C*. (**D**) In vivo tumor formation assay. HMLER cells expressing the corresponding ORFs were injected subcutaneously into immuno-compromised mice and tumor growth was monitored for 15 weeks. Shown are an image of tumor sizes (left) and quantification of tumor weights (right) (individual tumor growth curves are shown in *Figure 2—figure supplement 1E*). n = 9, **, p<0.01; *, p<0.05, Student's two-tailed t-test. (**E**) The percentage of CD44-high cells when SNAI1 is overexpressed along with shRNAs targeting candidate RNA-binding proteins or LacZ as a negative control. The red bar indicates a positive control shRNA against SNAI1 and the orange bars indicate shRNAs targeting *QKI* and *RBFOX1*. The knockdown efficiency is shown in *Figure 2—figure supplement 3D*. n = 4, ***, p<0.001; **, p<0.01; *, p<0.05, Student's two-tailed t-test. (**F–I**) The effect of CRISPR/Cas9-targeting of endogenous QKI or RBFOX1 in HMLE cells ectopically expressing SNAI1 on: (**F**) the percentage of CD44-high cells, (**G**) EMT marker expression (VIM: mesenchymal marker, CDH1: epithelial marker) quantified by qPCR, (**H**) EMT marker expression quantified by immunoblotting and (**I**) mammosphere formation. Also shown is the efficiency of

*Figure 2 continued on next page*

*Figure 2 continued*

gene knockout as quantified by immunoblotting. (B) n = 4, (C) n = 4, (E) n = 3, ***, p<0.001; **, p<0.01; *, p<0.05, Student's two-tailed t-test. (J) Summary of effects of ORFs on EMT markers and phenotypes.

DOI: https://doi.org/10.7554/eLife.37184.007

The following figure supplements are available for figure 2:

**Figure supplement 1.** Functional validation of candidate RNA-binding proteins in cellular assays.

DOI: https://doi.org/10.7554/eLife.37184.008

**Figure supplement 2.** QKI and RBFOX1 are regulated by EMT-TFs and promote tumor formation.

DOI: https://doi.org/10.7554/eLife.37184.009

**Figure supplement 3.** QKI and RBFOX1 promote an intermediate mesenchymal cell state.

DOI: https://doi.org/10.7554/eLife.37184.010

**Figure supplement 4.** The expression of QKI and RBFOX1 are altered in murine and human tumor samples that have undergone an EMT.

DOI: https://doi.org/10.7554/eLife.37184.011

Collectively, although QKI has been previously associated with AS changes occurring during EMT, our observations demonstrate that overexpression of QKI or RBFOX1 suffices to promote an intermediate mesenchymal and stem-like cell state and are also necessary for the SNAI1-induced EMT. Further, the expression of endogenous QKI and RBFOX1 were also induced by EMT-TFs such as SNAI1 or ZEB1 (*Figure 2J*). These results extend prior observations implicating these RNA binding proteins in EMT and confirm that our screen identified key regulators of EMT.

## Characterizing alternatively spliced transcripts that are regulated by QKI and RBFOX1

Although QKI and RBFOX2 (a homolog of RBFOX1) have been shown to regulate AS events during EMT (*Braeutigam et al., 2014*; *Venables et al., 2013*; *Yang et al., 2016*), it remains unclear whether QKI and RBFOX1 alter splicing of genes directly involved in EMT or if the expression of these RNA binding proteins merely correlate with the mesenchymal cell state. To dissect the mechanism by which QKI and RBFOX1 induce the intermediate mesenchymal and stem-like cell states, we overexpressed each of these or a control protein (hcRED or EGFP) in HME cells and used RNA-sequencing to assess changes in transcriptional programs. We subsequently used replicate multivariate analysis of transcript splicing (rMATS) to individually quantify and analyze differences in AS events in HME cells expressing either the hcRED or EGFP control proteins versus QKI, RBFOX1 or SNAI1 (*Shen et al., 2014*). Indeed, HME cells that expressed QKI or RBFOX1 exhibited a > 5 fold increase in the number of alternatively spliced events compared to control cells that expressed hcRED (*Figure 3—figure supplement 1A* and *Figure 3—source datas 1* and *2*). Among all detected types of splicing events, the majority of splicing changes after overexpression of QKI or RBFOX1 occurred in skipped exons (*Figure 3A*). We next used pre-ranked GSEA to analyze the pathways that are regulated by QKI or RBFOX1 and found that their downstream splicing targets were enriched in gene modules involved in cell motility/cytoskeleton organization, stem cell fate determination, oncogenic signaling and epigenetic targets (*Figure 3—figure supplement 1B,C*).

We then individually validated the top alternatively spliced genes regulated by both QKI and RBFOX1 with the hypothesis that shared targets were more likely to be involved in EMT. We focused on the genes with alternatively skipped exons, as it is the most prevalent type of AS in higher eukaryotes (*Keren et al., 2010*). We confirmed that, consistent with the RNA-seq results, pre-mRNAs of specific exons in the genes involved in cell motility/cytoskeleton organization, *FLNB, SLK, NUMB, CA12, ESYT2* and *ATP5C1* showed substantially greater skipping in cells expressing QKI and RBFOX1, as compared to control cells expressing hcRED or EGFP (*Figure 3B*). Interestingly, the same AS pattern for these genes was also observed in mesenchymal HME cells overexpressing SNAI1, indicating that the AS events observed in SNAI1-expressing cells are likely to be due to the activity of QKI and RBFOX1.

## QKI and RBFOX1 regulate the alternative splicing of FLNB

From our RNA-sequencing analysis, we found that many AS events, and in particular, skipped exons, were regulated by both QKI and RBFOX1 (*Figure 4A*, *Figure 4—figure supplement 1A* and *Figure 4—source data 1*). To identify direct targets of QKI and RBFOX1, we performed enhanced UV

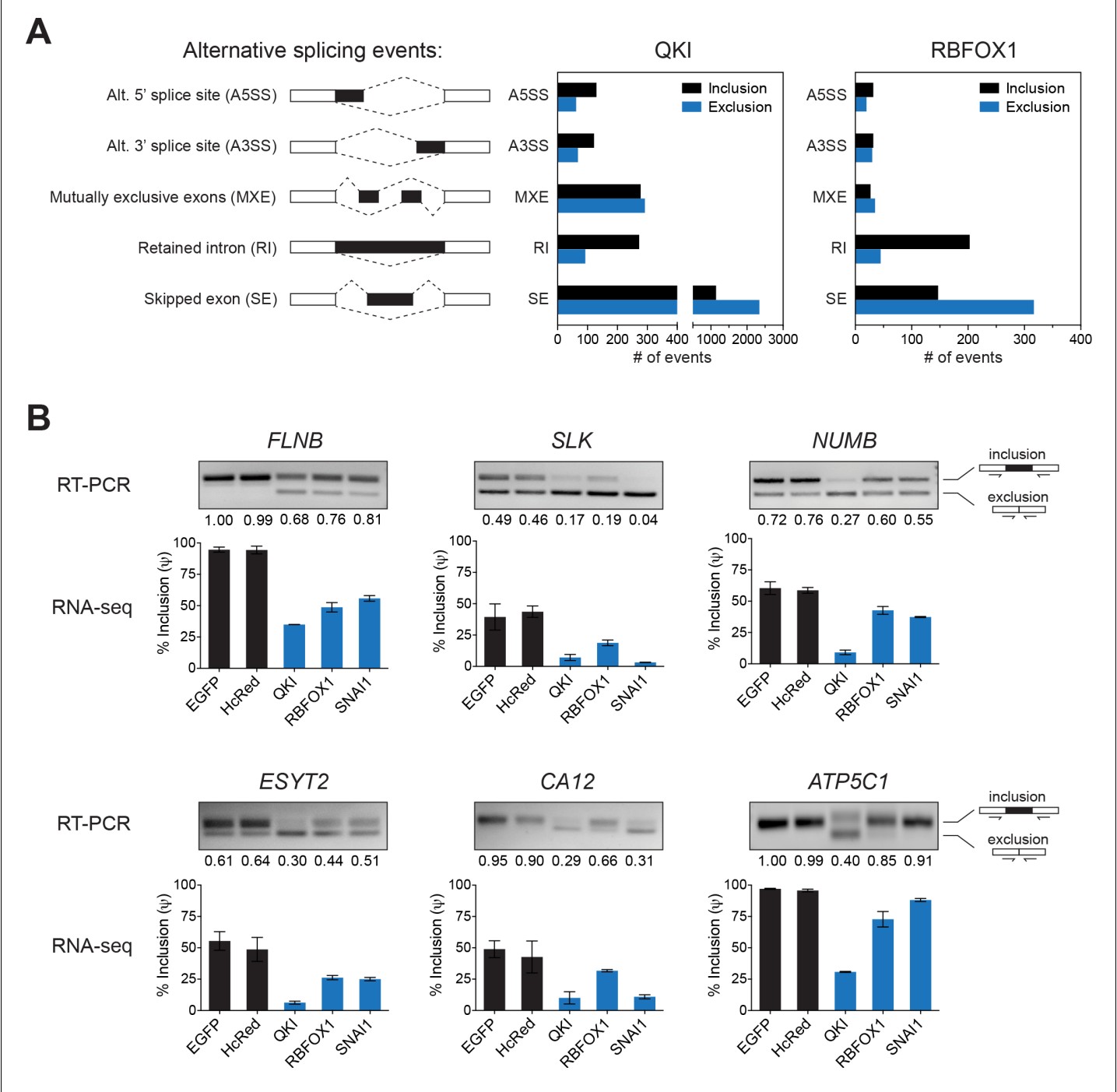

**Figure 3.** Identification of splicing targets regulated by QKI and RBFOX1. (**A**) Quantification of the different types of alternative splicing events regulated by QKI or RBFOX1 overexpression as determined using rMATS. Exclusion or inclusion are relative to control cells overexpressing EGFP. (**B**) RT-PCR validation of individual splicing events regulated by QKI or RBFOX1. The cDNA from cells expressing the indicated ORFs were subjected to PCR amplification using primers flanking the alternative exon. The ratios of the intensity of the upper (inclusion) and lower (exclusion) PCR product bands were quantified and the relative intensity of the upper band is indicated. Below are shown RNA-sequencing based quantification of the % inclusion of the alternative exon. n = 3 (EGFP, HcRed and RBFOX1) or n = 2 (QKI and SNAI1).

DOI: https://doi.org/10.7554/eLife.37184.012

The following source data and figure supplement are available for figure 3:

**Source data 1.** QKI rMATS splicing output.
DOI: https://doi.org/10.7554/eLife.37184.014

*Figure 3 continued on next page*

*Figure 3 continued*

**Source data 2.** RBFOX1 rMATS splicing output.
DOI: https://doi.org/10.7554/eLife.37184.015
**Figure supplement 1.** RNA sequencing analysis of HME cells expressing QKI and RBFOX1.
DOI: https://doi.org/10.7554/eLife.37184.013

crosslinking and immunoprecipitation followed by sequencing (eCLIP-seq) in HME cells (*Figure 4—figure supplement 1B*) (*Van Nostrand et al., 2016*). QKI-binding sites were located predominantly in introns, while the majority of RBFOX1-binding sites were found both in introns and 3'UTRs, and consistent with prior studies. We also recovered the known QKI (ACUAAC) and RBFOX1 (UGCAUG) binding motifs (*Figure 4B,C*). Interestingly, we found that QKI-binding sites were also highly enriched for the RBFOX-binding motif, UGCAUG (*Figure 4B,C*) and overall, there was a substantial degree of overlap between QKI and RBFOX1 eCLIP-binding peaks (p<0.001, *Figure 4D* and *Figure 4—source data 2*). When examining the 183 exon skipping events that we found to be regulated by both QKI and RBFOX1, we detected binding sites for both QKI and RBFOX1 for 36 events, with peaks overlapping the exon itself or positioned in the flanking introns (*Figure 4E*). Since the QKI and RBFOX1 proteins have previously been shown to physically associate with one another (*Lim et al., 2006*), we then tested whether these two proteins were also interacting in HME cells. When we isolated endogenous QKI complexes by immunoprecipitation, we detected a robust interaction with RBFOX1 protein that did not require the presence of RNA (*Figure 4—figure supplement 1C*). Thus, these observations demonstrate that QKI and RBFOX1 interact in human mammary epithelial cells and suggest that they concurrently bind to and regulate the AS of common downstream targets.

To identify transcripts whose AS is likely to play a functional role in promoting EMT, we assessed which QKI and RBFOX1-regulated AS events were also associated with an EMT gene signature across a panel of breast cancer cell lines from the Cancer Cell Line Encyclopedia (CCLE) (*Barretina et al., 2012*). We examined the AS events in breast cancer cell lines that were ranked by their EMT gene signature score (*Byers et al., 2013*), using the *Information Coefficient* (IC), an information-theoretic measure (*Kim et al., 2016*), and an empirical permutation test for statistical significance of the top hits (*Barretina et al., 2012*). Among all the common targets of QKI and RBFOX1, we found that *CD44* (IC:0.857, p value:<6.59e-07) and *FLNB* (IC:0.848, p value:<6.59e-07) scored as the top two genes that most strongly associated with the EMT signature in breast cancer cell lines (*Figure 4F*). *CD44* and *FLNB* were also among the top genes regulated by both QKI and RBFOX1 in HME cells (*Figure 4G*). Prior work has demonstrated that AS of CD44 to produce the standard shorter isoform promoting EMT, and that CD44 splicing is not only a marker of the EMT state but also contributes to EMT (*Brown et al., 2011*). However, the functional importance of FLNB in EMT has not yet been characterized. Exon 30 of FLNB is skipped when QKI and RBFOX1 are overexpressed (*Figure 3B*), and we found that both QKI and RBFOX1 were strongly bound to the intron flanking this exon (*Figure 4H*, QKI peak p value = 2.2e-16; RBFOX1 peaks p value = 3.0e-7 and 1.8e-9). Although RBFOX1-binding downstream of an exon typically results in splicing enhancement (*Conboy, 2017*), we found that binding of RBFOX1 downstream of FLNB exon 30 instead results in splicing repression. Together these observations support the view that QKI and RBFOX1 coordinately regulate the AS of genes associated with EMT.

## Alternative splicing of FLNB is strongly associated with basal-like breast cancer

Based on gene expression analysis, prior studies stratified breast cancer cell lines into basal B, basal A and luminal clusters, among which, the basal B subtype expresses mesenchymal markers and displays a high degree of stem-like cell features (*Kao et al., 2009*; *Neve et al., 2006*). To identify the alternative transcripts that correlated with the basal B subtype of breast cancer, we analyzed alternatively spliced events in breast cancer cell lines included in the CCLE (*Barretina et al., 2012*). We found several targets of QKI and RBFOX1, including *FLNB*, *SLK*, *USO1*, *ENAH*, *ESYT2*, *NUMB* and *ARHGEF1*, to be among the most differentially spliced genes in basal B breast cancer cell lines (*Figure 5—figure supplement 1A*). Strikingly, we observed a bimodal distribution for the AS of FLNB (*Figure 5A*), in which the shorter mesenchymal FLNB isoform corresponding to a lower exon 30

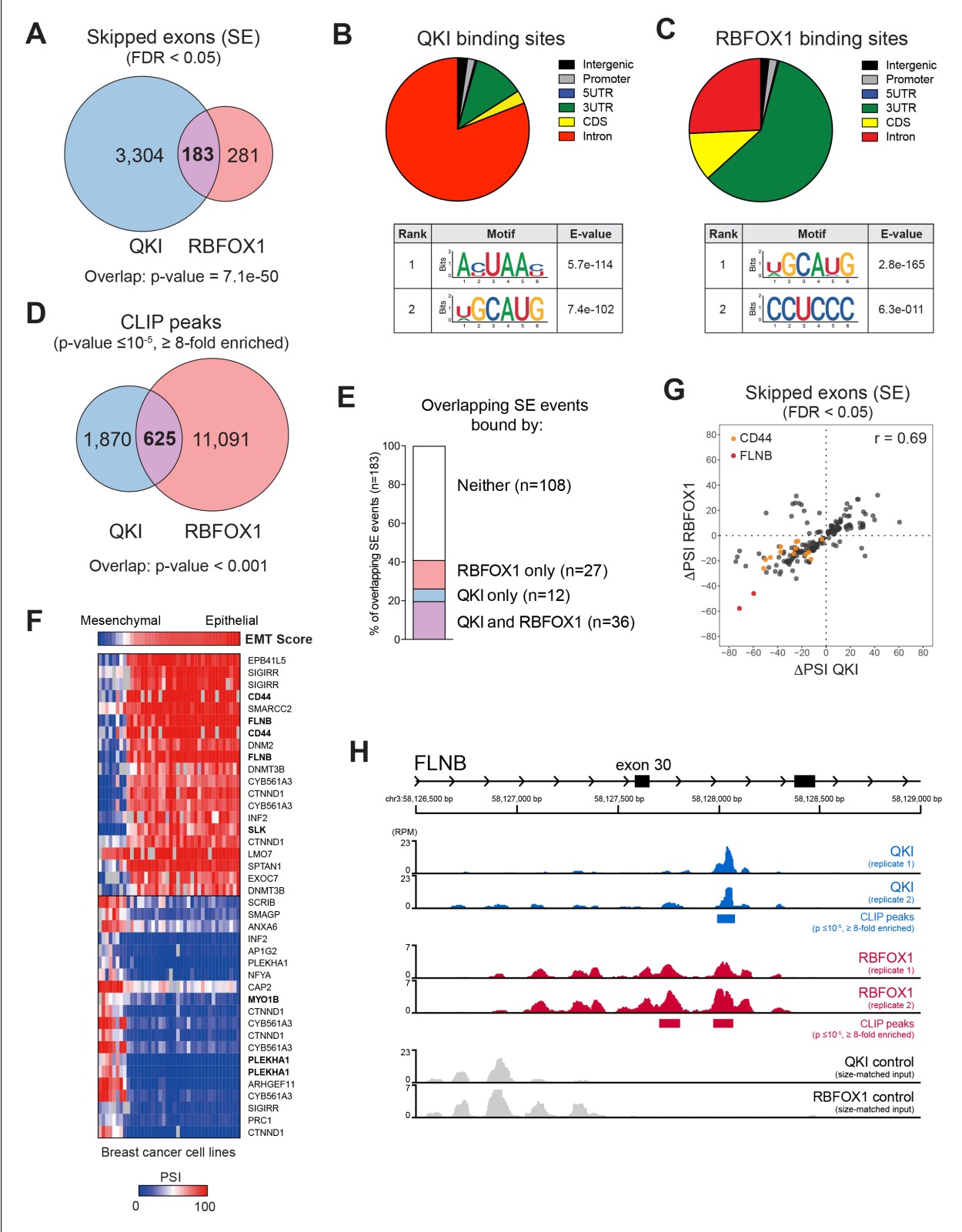

**Figure 4.** QKI and RBFOX1 cooperatively regulate the alternative splicing of common downstream targets, including FLNB. (**A**) Venn diagram illustrating the overlap in skipped exons regulated by QKI and RBFOX1, as detected by RNA-seq. The significance of overlap was determined by Fisher's exact test. (**B, C**) Binding distributions and de novo motifs identified from eCLIP-seq analysis of significant peaks (p≤10$^{-5}$, fold-enrichment ≥8). The QKI (**B**) or RBFOX1 (**C**) binding distributions are shown in pie charts (upper). Below are shown the top two ranked QKI (**B**) and RBFOX1 (**C**) binding

*Figure 4 continued on next page*

*Figure 4 continued*
motifs. (**D**) Venn diagram illustrating the overlap in binding peaks detected by eCLIP-seq analysis of QKI and RBFOX1. p<0.001, as determined by random shuffling of one set of peaks and repeated testing of the extent of overlap. (**E**) Percentage of SE splicing events regulated by both QKI and RBFOX1 (n = 183), that contain CLIP-binding peaks located in the exon or flanking introns, for either or both proteins. (**F**) Heatmap depicting the association between an EMT signature score (top row) and AS events in a panel of breast cancer cell lines. Shown are the top 20 AS events that are positive or negatively associated with the EMT score. (**G**) Scatter plot of the change in 'Percentage Spliced In' (PSI) for SE events shared between QKI and RBFOX1. ΔPSI values are for each ORF relative to EGFP. Colored dots indicate events in CD44 or FLNB. (**H**) Tracks indicate eCLIP-seq read density for QKI and RBFOX1 at FLNB exon 30, normalized to a million total usable reads. Below each pair of replicate tracks are shown significant binding peaks called by CLIPper software ($p{\leq}10^{-5}$, fold-enrichment $\geq$8). Also shown are size-matched input controls for each pair of replicates.
DOI: https://doi.org/10.7554/eLife.37184.016

The following source data and figure supplement are available for figure 4:

**Source data 1.** QKI and RBFOX1 overlapping events.
DOI: https://doi.org/10.7554/eLife.37184.018
**Source data 2.** QKI and RBFOX1 CLIP peaks.
DOI: https://doi.org/10.7554/eLife.37184.019
**Figure supplement 1.** RNA-Seq and eCLIP-Seq analysis of HME cells expressing QKI and RBFOX1.
DOI: https://doi.org/10.7554/eLife.37184.017

'Percent Spliced In' (PSI) value, occurred overwhelmingly in basal B cell lines, while the longer epithelial FLNB isoform existed predominantly in luminal and basal A cell lines. We validated this finding in two basal B (BT549 and MDAMB231) and two luminal (ZR75-1 and MCF7) cell lines by RT-PCR (*Figure 5A*).

We further found that there is a strong association between the AS of FLNB exon 30 and EMT gene expression features in breast cancer cell lines (*Figure 5—figure supplement 1B*). When we examined all non-hematopoietic cancer cell lines in the CCLE, we found that the degree of FLNB exon 30 splicing correlated significantly with a ZEB1 target signature, an epithelial differentiation signature, two metastasis signatures and a mammary stem cell signature (*Figure 5B* and *Figure 5—figure supplement 1C*). These observations further confirmed that the AS of FLNB exon 30 strongly associates with EMT and a stem-like cell state. In addition, the strong association between FLNB splicing and EMT features suggest that FLNB exon 30 splicing may serve as a biomarker for residence of cancer cells in a mesenchymal state.

Since mesenchymal and stem-like cell features are enriched in basal-like breast cancer, we examined whether the splicing of FLNB and the expression of QKI or RBFOX1 were associated with the basal-like subtype in TCGA Breast Invasive Carcinoma (BRCA) samples. We observed lower expression of the longer FLNB isoform with exon 30 included and higher expression of the shorter FLNB isoform in samples classified as the basal subtype, consistent with the notion that FLNB splicing plays a role in regulating the mesenchymal and stem-like cell state (*Figure 5—figure supplement 1D*). Similarly, we discovered elevated expression of QKI (NM_006775, also called QKI-5) in basal-like breast cancers relative to other subtypes of breast cancers (*Figure 5—figure supplement 1E*).

## FLNB isoform switching promotes a mesenchymal-like cell state

FLNB is a member of the Filamin family of actin-binding proteins (FLNA, B and C). Prior work has implicated the role of filamins in actin crosslinking, focal adhesion kinase and integrin signaling, and regulating transcriptional activity (*Feng and Walsh, 2004*; *van der Flier et al., 2002*; *Xu et al., 2010*; *Zhou et al., 2010*). Filamins share an N-terminal actin-binding domain, two hinge regions, and 24 filamin-type immunoglobulin-like (FLN) domains that are involved in the formation of tail-to-tail dimers (*Feng and Walsh, 2004*; *van der Flier et al., 2002*). Exon 30 of FLNB encodes the first hinge (H1) domain, which governs filamin protein flexibility and calpain cleavage sensitivity (*Figure 6A*) (*Feng and Walsh, 2004*; *Xu et al., 2010*). The skipping of exon 30 results in loss of the H1 domain from the full-length protein without altering the remainder of the protein. Hereafter, we refer to the longer isoform of FLNB (including exon 30) as FLNB-L, and to the shorter isoform (which lacks exon 30) as FLNB-ΔH1. When we tested whether the splicing of FLNB differed between the CD44-high and CD44-low cell populations (*Figure 6B*), we found that FLNB exon 30 skipping occurred exclusively in the CD44-high mesenchymal and stem-like cell population.

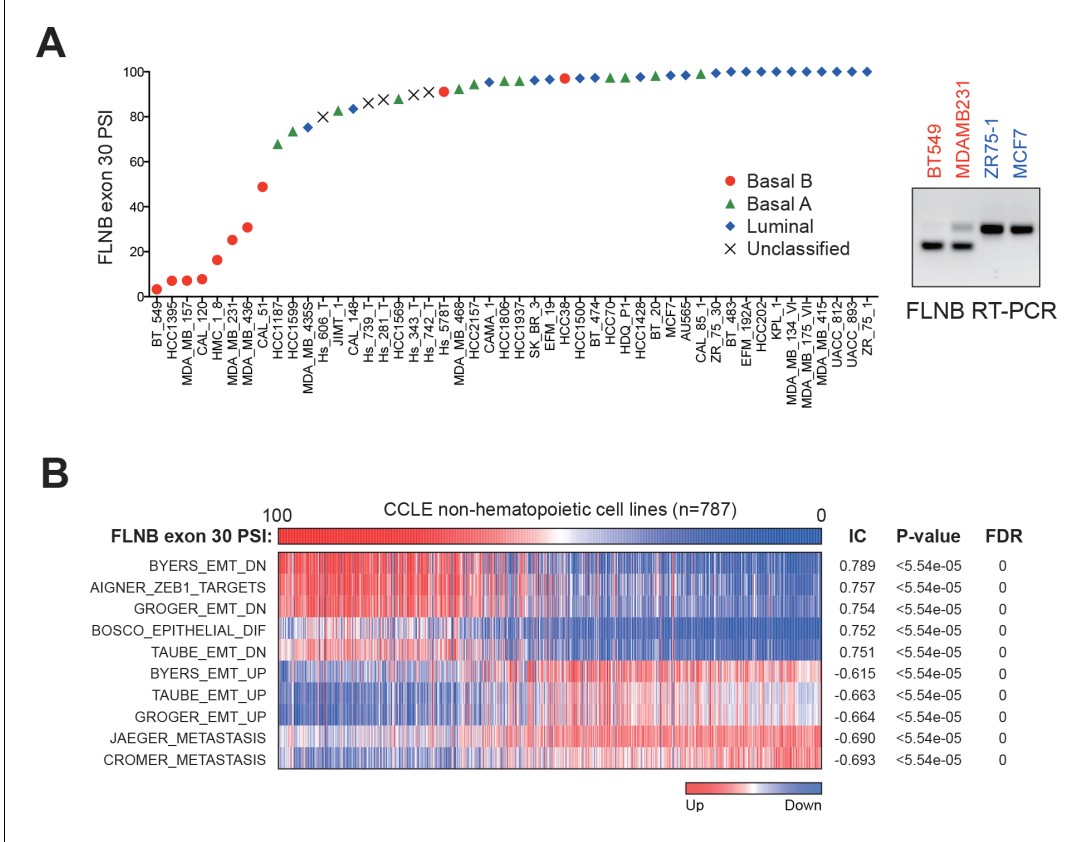

**Figure 5.** Alternative splicing of FLNB exon 30 is strongly associated with basal-like breast cancer. (**A**) FLNB exon 30 PSI in all CCLE breast cancer cell lines. RT-PCR validation of FLNB exon 30 splicing in two Basal B cell lines (BT549 and MDAMB231) and two luminal cell lines (ZR75-1 and MCF7) is shown on the right. (**B**) CCLE cell lines (except hematopoietic cell lines, n = 787) were ranked by their FLNB exon 30 Percent Spliced In (PSI) values, and associated gene expression signatures were analyzed by the REVEALER program. The top associated signatures with their Information coefficient (IC), p values and False Discovery Rates (FDR) are shown.

DOI: https://doi.org/10.7554/eLife.37184.020

The following figure supplement is available for figure 5:

**Figure supplement 1.** Alternative splicing of FLNB exon 30 and the expression of QKI correlate with EMT-related gene signatures in cancer cell lines and TCGA breast cancer patient samples.

DOI: https://doi.org/10.7554/eLife.37184.021

To investigate the function of FLNB in regulating EMT, we suppressed FLNB expression using siRNAs targeting the *FLNB* 3'UTR region in HMLE cells, in which the FLNB-L isoform represents the majority of FLNB protein. We found that suppression of FLNB-L upregulated the expression of mesenchymal markers, VIM and FN1, indicating that FLNB-L plays a negative role in regulating EMT (*Figure 6C*). To dissect the respective role of FLNB-L and FLNB-ΔH1 isoforms, we rescued the suppression of endogenous FLNB by ectopically expressing each isoform of FLNB (*Figure 6D*). Depletion of FLNB promoted the expression of mesenchymal markers. We found that FLNB-L reduced the expression of mesenchymal markers, FN1 and VIM. In contrast, the expression of FLNB-ΔH1 did not decrease the mesenchymal marker expression. When the two isoforms of FLNB were expressed in a mesenchymal cell line, MDA-MB-231, we also found that FLNB-L overexpression suppressed mesenchymal marker expression, strongly suggesting that FLNB-L inhibits the EMT (*Figure 6—figure supplement 1A*). Interestingly, when we expressed each isoform in HMLE cells in the presence of the endogenous FLNB-L, FLNB-ΔH1 ectopic expression elevated mesenchymal markers while the expression of FLNB-L did not significantly alter expression of the same set of markers (*Figure 6—figure supplement 1B,C*). Since filamin proteins dimerize (*Berry et al., 2005*; *Pudas et al., 2005*; *Stossel et al., 2001*), the effects of FLNB-ΔH1 proteins likely represent interactions with the endogenous FLNB-L, which blocks the suppressive effect mediated by FLNB-L. As before, we did not

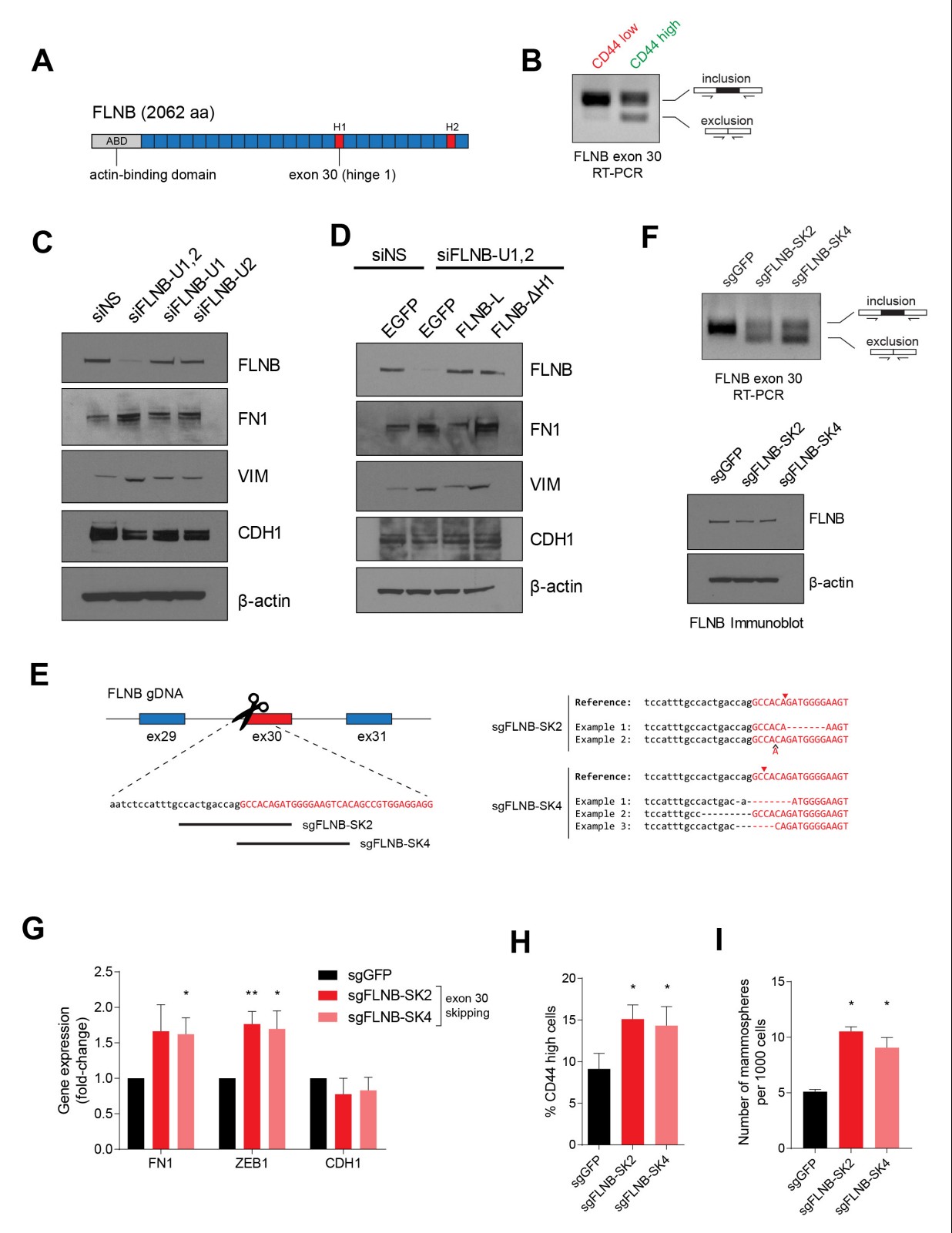

**Figure 6.** FLNB isoform switching promotes the mesenchymal cell state. (**A**) Schematic of FLNB (Filamin B) protein domain structure, which contains an N-terminal actin-binding domain (ABD, shown in yellow), 24 filamin repeats (shown in blue) and two hinge domains (H1 and H2, shown in red). The exon 30 of FLNB encodes the first hinge domain (**H1**). (**B**) CD44-high and CD44-low HMLER cells were sorted and FLNB AS was analyzed by RT-PCR with primers flanking FLNB exon 30. The CD44-high HMLER cells were spontaneously generated. (**C**) The expression of FLNB was suppressed by

*Figure 6 continued on next page*

*Figure 6 continued*

siRNAs against FLNB 3'UTR region (FLNB-U1, U2 and UTR1,2) in HMLE cells. FLNB-U1,2 is a pool of FLNB-U1 and FLNB-U2 siRNAs. Protein levels of FN1, VIM (mesenchymal markers), CDH1 (epithelial marker) and FLNB after FLNB suppression was quantified by immunoblotting. (D) The expression of FLNB was rescued by FLNB-L (FLNB long isoform) or FLNB-ΔH1 (FLNB lacking H1 domain) after endogenous FLNB depletion by siRNA targeting the UTR of FLNB (FLNB-U1,2). Protein levels of FN1, VIM (mesenchymal markers), CDH1 (epithelial marker) and FLNB were quantified by immunoblotting. (E) Schematic of CRISPR/Cas9-mediated targeting of the FLNB intron 29 - exon 30 junction. (Left) Guide RNAs were designed to target the junction of FLNB intron 29 (in lower-case black letters) and exon 30 (in capital red letters). (Right) Examples of modifications to the junction induced by sgRNAs sgFLNB-SK2 and sgFLNB-SK4 are shown. (F) FLNB exon 30 skipping and total FLNB levels were quantified in cells expressing sgRNAs targeting the junction of FLNB intron 29 and exon 30 by RT-PCR (upper) and immunoblot (lower) (corresponding to *Figure 6E*). (G–I) Effect of CRISPR/Cas9-mediated modulation of the FLNB-L/FLNB-ΔH1 ratio on the expression of mesenchymal and epithelial markers as examined by qPCR (G), on the expression of CD44 as analyzed by flow cytometry (H), and on the number of mammosphere formed per 1000 cells cultured in low attachment condition (I). n = 4 for (G), n = 6 for (H), n = 3 for (I). *, p<0.05; **, p<0.01, Student's two-tailed t-test compared to sgGFP controls.
DOI: https://doi.org/10.7554/eLife.37184.022
The following figure supplement is available for figure 6:

**Figure supplement 1.** FLNB regulates EMT.
DOI: https://doi.org/10.7554/eLife.37184.023

observe robust changes in the expression of pre-existing epithelial markers in these experiments, reminiscent of our previous observation that QKI or RBFOX1 induces an intermediate mesenchymal state with retention of epithelial markers and acquisition of mesenchymal ones (*Figure 2A,B* and *Figure 2—figure supplement 1B*). Together, these results support the view that the skipping of the exon 30 of FLNB switches the function of the FLNB from suppressing to promoting the EMT.

To manipulate the ratio of the two FLNB isoforms and dissect the function of the FLNB exon 30 skipping, we modified the genomic locus of the intron-exon junction using CRISPR/Cas9-mediated genome editing to skew the isoform ratio of the endogenous FLNB transcripts. We designed several sgRNAs that target the junction of intron 29 and exon 30 (sgFLNB-SK2 and SK4). Remarkably, we found that disrupting this junction in the genomic locus was effective in causing skipping of the endogenous exon 30 of FLNB (*Figure 6E,F*). In line with our previous observations, FLNB exon 30 skipping induced by this approach also increased the expression of mesenchymal markers (*Figure 6G*). We also discovered that FLNB exon 30 skipping induced a modest but significant increase in the CD44-high cell population and in the number of mammospheres under low attachment growth conditions (*Figure 6H,I*).

In addition, we used two sets of siRNAs that targeted either exon 30, or the junction between exon 29 and exon 31 when exon 30 is skipped. The siRNAs that target the exon 29–31 junction selectively disrupt formation of the FLNB-ΔH1 isoform, since the FLNB-L isoform lacks the siRNA target sequences. This approach effectively altered the ratio of FLNB-L and FLNB-ΔH1 (*Figure 6—figure supplement 1D,E*) and revealed that an elevated ratio of the FLNB-ΔH1 isoform over the FLNB-L isoform and a reduction of FLNB protein levels significantly increased the level of mesenchymal markers, consistent with the effect of ectopically expressing FLNB-ΔH1 or CRISPR/Cas9-mediated editing of the splice junction (*Figure 6C–I*). Together, these observations demonstrate that the skipping of FLNB exon 30 contributes to the acquisition of a mesenchymal-like cell state.

## EMT mediated by FLNB exon 30 skipping is dependent on the FOXC1 transcription factor

In addition to their function in the cytoplasm, actin-binding proteins, such as the Filamin family, have been shown to localize to the nucleus and regulate transcription and gene expression (*Bedolla et al., 2009*; *Berry et al., 2005*; *Olson and Nordheim, 2010*; *Zheng et al., 2009*; *Zhou et al., 2010*). We tested whether the two isoforms of FLNB generated by alternative splicing of exon 30 localized to different subcellular compartments. By comparing HMLE cells expressing a control sgRNA targeting GFP (sgGFP) or sgRNAs that induce exon 30 skipping (sgFLNB-SK2 and SK4) (*Figure 6E*), we found that the FLNB-L isoform localized to both the cytoplasm and the nucleus while FLNB-ΔH1 was preferentially localized to the cytoplasm (50% reduction in nuclear localization, *Figure 7A*). We confirmed this change of FLNB localization using immunofluorescence (*Figure 7—figure supplement 1A*). Thus, alternative splicing of exon 30 changes the nuclear localization of FLNB.

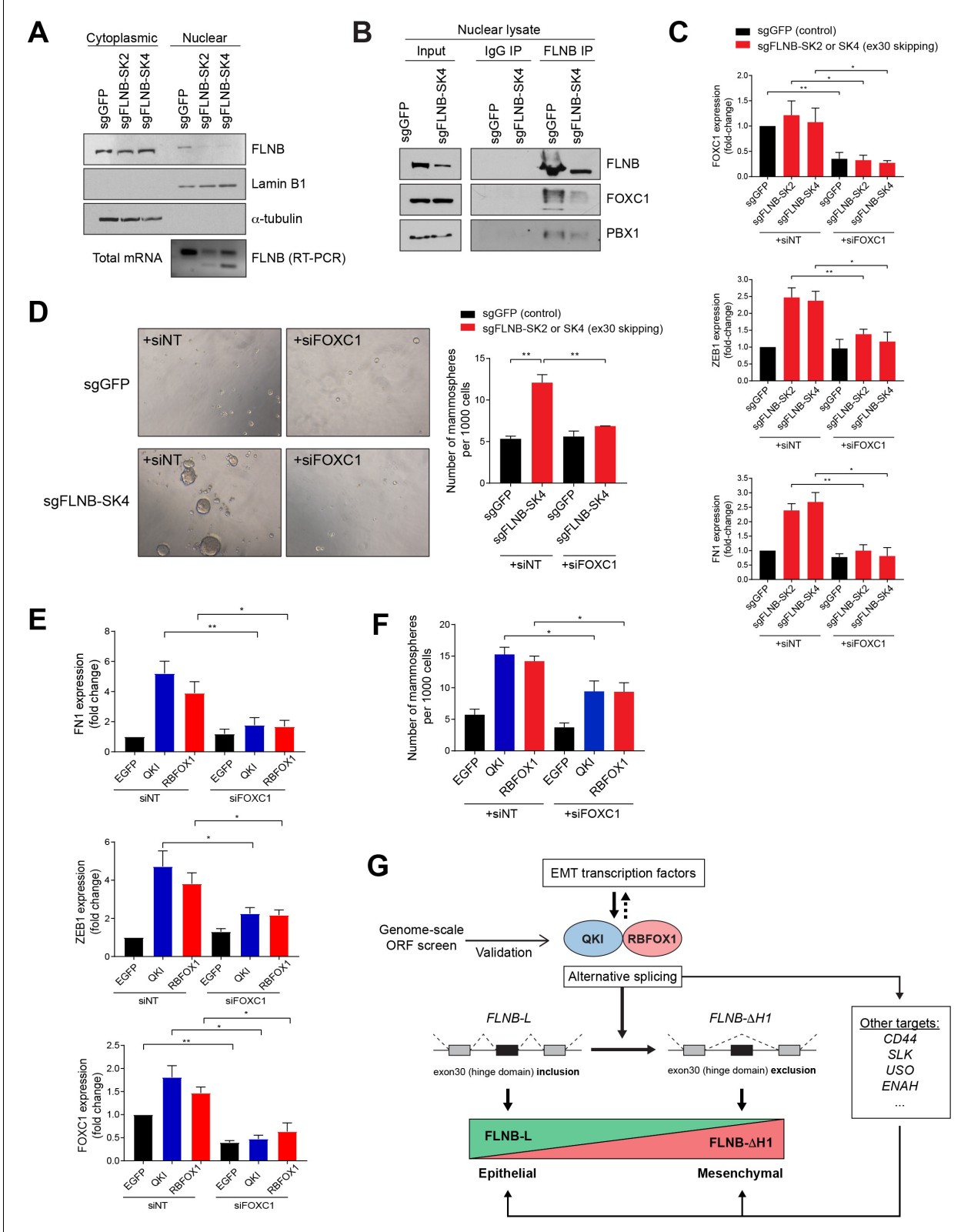

**Figure 7.** EMT mediated by FLNB isoform switching is dependent on the FOXC1 transcription factor. (**A**) Analysis of nuclear and cytoplasmic fractions of FLNB. HMLE cells expressing Cas9 and sgRNAs targeting GFP (sgGFP, as control) or targeting the FLNB intron 29 - exon 30 junction to induce exon skipping (sgFLNB-SK2 and SK4, described in *Figure 6E*), were subjected to subcellular fractionation. The levels of FLNB proteins in the nuclear and cytoplasmic fractions were analyzed by immunoblotting and the skipping of exon 30 was confirmed by RT-PCR. The nuclear levels of FLNB is

*Figure 7 continued on next page*

*Figure 7 continued*

diminished when exon 30 is skipped. (**B**) Interaction between FLNB, FOXC1 and PBX1 in the nucleus as determined by immunoprecipitation followed by immunoblotting. HMLE cells expressing Cas9 and sgRNAs targeting GFP (sgGFP, as control) or targeting the FLNB intron 29 - exon 30 junction to induce exon skipping (sgFLNB-SK4, described in *Figure 6E*), were subjected to immunoprecipitation analysis using nuclear protein extract. The interaction between FLNB and FOXC1 in the nuclear extract was reduced largely due to the decrease in nuclear FLNB levels. (**C**) The depletion of FOXC1 diminishes the upregulation of mesenchymal markers (ZEB1 and FN1) mediated by FLNB exon 30 skipping (sgFLNB-SK2 and SK4, shown in red bars) as measured by qPCR. n = 6, *, p<0.05; **, p<0.01, Student's two-tailed t-test. The knockdown efficiency for FOXC1 is shown in the upper panel. (**D**) The rate of mammosphere formation in HMLE cells that expressed the indicated sgRNAs and transfected with a FOXC1 siRNA pool. The number of mammospheres per 1000 cells plated was counted. Representative images (left) and quantification (right) are shown. n = 3, **, p<0.01, Student's two-tailed t-test. (**E, F**) Depletion of FOXC1 diminishes the upregulation of mesenchymal markers (**E**) and mammosphere formation (**F**) mediated by QKI and RBFOX1 expression, as measured by qPCR (**E**) and mammosphere formation assay (**F**). n = 6 for (**E**), n = 4 for (**F**). *, p<0.05; **, p<0.01, Student's two-tailed t-test. (**G**) A model of the mechanism underlying the QKI and RBFOX1-induced mesenchymal and stem-like cell state.
DOI: https://doi.org/10.7554/eLife.37184.024

The following figure supplement is available for figure 7:

**Figure supplement 1.** EMT mediated by FLNB isoform switching is dependent on the FOXC1 transcription factor.
DOI: https://doi.org/10.7554/eLife.37184.025

Filamin A has been reported to assemble a protein complex with FOXC1 and PBX1, which inhibits FOXC1 transcriptional activity (*Berry et al., 2005*; *Zheng et al., 2009*; *Zhou et al., 2010*). To test whether Filamin B also forms a complex with FOXC1 in HMLE cells, we confirmed that FLNB interacts with FOXC1 by co-immunoprecipitation (*Figure 7—figure supplement 1B*). Furthermore, we found that the interaction among FLNB, FOXC1 and PBX1 was reduced when we induced FLNB exon 30 skipping, largely due to the decreased amount of nuclear FLNB protein (*Figure 7B*). Based on these observations, we conclude that FLNB nuclear exclusion, mediated by exon 30 skipping, regulates its interaction with FOXC1.

FOXC1 is a transcription factor that induces EMT (*Han et al., 2017*; *Huang et al., 2017*; *Ou-Yang et al., 2015*; *Zhu et al., 2017*). Since the nuclear filamins inhibit FOXC1 activity (*Berry et al., 2005*; *Zheng et al., 2009*; *Zhou et al., 2010*), we hypothesized that the reduced nuclear localization of FLNB promotes EMT by releasing FOXC1 from FLNB. Specifically, we tested whether the EMT induced by FLNB isoform switching was dependent on FOXC1. Indeed, we found that FOXC1 depletion by siRNA significantly dampened the upregulation of mesenchymal marker expression (*Figure 7C*) and the formation of mammospheres (*Figure 7D*) mediated by FLNB exon 30 skipping. Furthermore, we found that FOXC1 is also partially required for the upregulation of mesenchymal marker expression (*Figure 7E*) and increase in mammosphere formation (*Figure 7F*, *Figure 7—figure supplement 1C*) induced by QKI and RBFOX1 expression, which regulate the alternative splicing of FLNB exon 30. In summary, the skipping of FLNB exon 30 promotes EMT by reducing FLNB nuclear localization and release of the FOXC1 transcription factor.

## Discussion

Splicing is a key step in the regulation of almost all human transcripts. The recent genomic characterization of cancers has revealed recurrent somatic mutations and copy number alterations in RNA splicing factors and RBPs in a significant subset of human tumors (*Dvinge et al., 2016*). Cancer cells harboring aberrant splicing factor expression or mutations in genes encoding splicing factors display unique cancer-specific mis-splicing that may facilitate tumor formation and progression. Although alternative splicing has been associated with EMT previously, in-depth studies are needed to better understand the mesenchymal cell state-specific RBPs and their functional downstream targets. Here, we found that QKI and RBFOX1 regulate the splicing of an exon in the actin-binding protein FLNB to regulate the EMT in breast cancer. This finding suggests that the AS of a single exon may serve as a highly quantifiable surrogate molecular biomarker for the process of EMT in solid tumors.

Recent work has shown that cells often progress through a spectrum of intermediate states between the fully epithelial and fully mesenchymal cell phenotypes (*George et al., 2017*; *Nieto et al., 2016*). We found that the mesenchymal cell state mediated by the expression of QKI and RBFOX1 exhibited upregulation of mesenchymal markers with continued retention of certain epithelial markers, indicating that this cell state is one that lies in-between the fully epithelial and

fully mesenchymal poles of this spectrum. Consistent with prior studies (*Nieto et al., 2016*; *Schmidt et al., 2015*), our results suggest that the intermediate/partial mesenchymal cell state displays a high degree of stem cell features and fosters tumor formation in vivo.

The control of AS by RNA-binding proteins is highly context dependent (*Fu and Ares, 2014*) and tissue specific (*Yeo et al., 2004*). QKI has been shown to be a tumor suppressor in brain tumors (*Chen et al., 2012*), colon cancers (*Taube et al., 2010*) and prostate cancers (*Zhao et al., 2014*). In contrast, QKI has also been reported to promote tumor formation in both esophageal carcinoma (*He et al., 2016*) and glioblastoma (*Wang et al., 2013*; *Xi et al., 2017*). Here, we found that QKI promoted tumor formation in human mammary epithelial cells. These distinct observations may be due to the differences in the initial cell states in which the cancer cells may reside.

The RBFOX1 ORF that we isolated from the screen encodes an isoform (NM_145893.2) that has been previously shown to partially localize to the cytoplasm in neuronal cells (*Lee et al., 2009*). In breast cells, we observed that 38% of this RBFOX1 isoform (NM_145893.2) localize in the nucleus to regulate pre-mRNA splicing (*Figure 7—figure supplement 1D*). Further studies will be needed to determine whether the cytoplasmic fraction of RBFOX1 also plays an additional role in regulating EMT.

Interestingly, the overexpression of EMT-related TFs such as SNAI1 and ZEB1 induced the QKI- and RBFOX1-mediated splicing program. Moreover, overexpression of the QKI and RBFOX1 splicing factors themselves promoted a mesenchymal cell state in which SNAI1 and ZEB1 expression were also elevated. These observations indicate that transcriptional and post-transcriptional regulation of EMT complement and regulate one another, suggesting how EMT can be dynamically controlled.

Filamins bind to proteins with diverse functions and important roles in multiple cellular process such as the regulation of cell signaling, transcription and organ development (*Zhou et al., 2010*). In this study, we demonstrated that the skipping of exon 30 in FLNB, which encodes a hinge region (H1), not only serves as a marker for mesenchymal cells but also promotes EMT by releasing the FOXC1 transcription factor from an inhibitory complex. These observations provide a mechanistic explanation of how mesenchymal-specific splicing factors such as QKI and RBFOX1 induce EMT.

Collectively, we conclude that QKI and RBFOX proteins play important roles in establishing the mesenchymal and stem-like cell state in breast cancers, which is in part mediated through their mutual regulation of the skipping of FLNB exon 30. Alternative splicing of pre-mRNAs represents a mechanism for flexibly regulating gene expression by enabling the generation of protein isoforms with distinct or even opposing functions without altering rates of transcription. As such, it offers yet another level of epigenetic control of gene expression. Accordingly, it may provide another means of regulation that tumor cells exploit in order to produce proteins that favor cell survival and cell state changes, such as the EMT programs studied here.

## Materials and methods

**Key resources table**

| Reagent type or resource | Designation | Source or reference | Identifiers | Additional information |
|---|---|---|---|---|
| Antibody | anti-Pan-QKI | Millipore | MABN624 | |
| Antibody | anti-QKI | Bethyl | A300-183A RRID:AB_2173160 | |
| Antibody | anti-RBFOX1 | Sigma | SAB2100002 RRID:AB_10600323 | |
| Antibody | anti-MBNL1 | Santa-Cruz | sc-47740 RRID:AB_784435 | |
| Antibody | anti-MBNL2 | Santa-Cruz | sc-136167 RRID:AB_2140469 | |
| Antibody | anti-CELF4(Brunol4) | Santa-Cruz | sc-398292 | |
| Antibody | anti-RBFOX2 (RBM9) | Bethyl | A300-864A RRID:AB_609476 | |
| Antibody | anti-FLNB | Cell signaling | 12979S | |

*Continued on next page*

*Continued*

| Reagent type or resource | Designation | Source or reference | Identifiers | Additional information |
|---|---|---|---|---|
| Antibody | anti-SNAI1 | Cell signaling | 3879S<br>RRID:AB_2255011 | |
| Antibody | anti-Vinculin | Cell signaling | 4650S<br>RRID:AB_10559207 | |
| Antibody | anti-V5 | Cell signaling | 13202 | |
| Antibody | anti-V5 (for eCLIP) | Life Technologies | R96025<br>RRID:AB_159313 | |
| Antibody | anti-CD44-PE-Cy7 | Affymetrix | 25-0441-81<br>RRID:AB_469622 | |
| Chemical compound, drug | siRNA NS #1 | GE Dharmacon | D-001810–01- 05 | |
| Chemical compound, drug | siFLNB-L1 (3019) | GE Dharmacon | CTM-237650/<br>LJBYN-000019 | |
| Chemical compound, drug | siFLNB-L2 (3021) | GE Dharmacon | CTM-237646/<br>LJBYN-000021 | |
| Chemical compound, drug | siFLNB-S1 (2923) | GE Dharmacon | CTM-237648/<br>LJBYN-000023 | |
| Chemical compound, drug | siFLNB-S2 (2925) | GE Dharmacon | CTM-237649/<br>LJBYN-000025 | |
| Chemical compound, drug | siFOXC1 smart pool of 4 siRNAs | GE Dharmacon | M-009318-01-0005 | |
| Chemical compound, drug | siFLNB-U1 (targeting UTR region of FLNB) | GE Dharmacon | CTM-206669 | |
| Chemical compound, drug | siFLNB-U2 (targeting UTR region of FLNB) | GE Dharmacon | CTM-206671 | |
| Chemical compound, drug | Lipofectamine RNAiMAX Transfection Reagent | thermofisher | 13778150 | |
| Commercial assay or kit | QIAamp DNA Mini Kit | Qiagen | 51304 | |
| Commercial assay or kit | RNeasy Mini kit | Qiagen | 74104 | |
| Gene | QKI | NCBI Reference | NM_006775.2 | |
| Gene | RBFOX1 | NCBI Reference | NM_145893.2 | |
| Strain, strain background | NCr-nude mice | Taconic | NCRNU,F,<br>CrTac:NCr,Foxn1nu | |

## Cell culture

The human mammary epithelial (HME) cell line, and its derived cell lines HMLE and HMLER, were grown in Mammary Epithelial Cell Growth Medium (MEGM, Lonza, #CC-3150). MCF7, BT549, MDAMB231, ZR75-1 cells were grown in Dulbecco's minimum essential medium (DMEM) or Roswell Park Memorial Institute (RPMI) medium containing 10% fetal bovine serum and 1% antibiotics, while 293 T cells were grown in DMEM supplemented with 10% fetal bovine serum and 1% antibiotics. All cell lines were obtained from the Cancer Cell Line Encyclopedia directly from original sources and had their identity confirmed by SNP fingerprinting (*Barretina et al., 2012*). All cells were tested for mycoplasma periodically.

## Plasmids and ORFeome library

The ORFs of EGFP, HcRed, CELF4, MBNL1, MBNL2, QKI, RBFOX1, SFPQ, HNRNPUL1, SRSF9, RBM47, TGFBR2 and SNAI1 were obtained from the Genetic Perturbation Platform at the Broad Institute. Plasmids containing the two isoforms of FLNB (FLNB-L and FLNB-ΔH1) were generous gifts from Dr. Arnoud Sonnenberg at the Netherlands Cancer Institute. The ORFeome 8.1 library was produced in the Genetic Perturbation Platform at the Broad Institute. There are the 17,255 ORF clones

in the library, among which, 12952 ORF clones have at least 99% nucleotide and protein match (75.1%). Of those genes 7547 ORF clones are 100% protein matches (43.7%) and 6040 are 100% nucleotide matches (35.0%). The construction of the ORF library has been described previously (*Yang et al., 2011*).

## Animal studies

All protocols with use of animals were approved by DFCI's Institutional Animal Care and Use Committee (IACUC). For tumor studies, HMLER cells expressing different ORFs were washed and suspended in 50% Matrigel/PBS mix (Corning Matrigel Basement Membrane Matrix, #354234), then injected subcutaneously in both flanks and in the back of 6-week-old female immunocompromised NCr-nude mice (Taconic, NCRNU-F, CrTac:NCr-Foxn1nu). Two million cells were injected per site. Mice were sacrificed after 15 weeks or when tumors reached a diameter of 2 cm.

## Flow cytometry

Cells were trypsinized, suspended in phosphate-buffered saline (PBS) with CD44-PE-Cy7 antibody (Affymetrix # 25-0441-81; 1:500 dilution), and stained for 20 min at room temperature; cells were mixed at 5 min intervals, and then washed with PBS to remove excess antibodies. Immediately after, cells were sorted on a BD FACSAria SORP or analyzed on a BD Fortessa, using BD FACSDiva Software (BD Biosciences, USA).

## Mammosphere assay

Mammosphere cultures were generated as described (*Chaffer et al., 2013*). Briefly, 1000 cells were seeded per well in a 96-well Corning Ultra-Low attachment plate, in replicates of 6 (Corning, USA; CLS3474). Cells were grown in a serum-free mammary epithelial cell growth medium, supplemented with B27 (Invitrogen), 10 ng/mL EGF, 20 ng/mL bFGF (BD Biosciences) and 1% methycellulose. Bovine pituitary extract was excluded. Spheroid numbers were counted between days 8 and 12 microscopically.

## ORF screening

The genome-scale screen was performed in two biological replicates. HMLER cells in the CD44-low state were pre-sorted using flow cytometry. The purity of CD44-low cells was >99.99% for each experiment. HMLER_CD44-low cells were then transduced with the ORF library and cultured for 7 days, with one passage on Day 4. On Day 7, CD44-high cells were sorted from more than 200 million transduced HMLER cells, using flow cytometry for each biological replicate. Sorted CD44-high cells (about 200 thousand cells for each replicate) and their corresponding unsorted HMLER cells were subjected to genomic DNA extraction using the QIAamp DNA Mini Kit (Qiagen # 51304). The barcodes corresponding to each ORF were amplified using PCR, and analyzed by next-generation sequencing. Enriched barcodes were analyzed as follows: (i) Each sample was normalized to a total of 1 million barcode reads. (ii) The number of each barcode after normalization was calculated to its log base two value. The log value of each barcode in the unsorted group was subtracted from the CD44-high group to obtain the log fold-change in the value of each barcode. (iii) The averages and standard deviations (SD) of the log fold-change values in all samples were determined, and Z scores for each barcode were calculated as follows: $Z_{Barcode\ X}$ = (Log value $_{Barcode\ X}$ - average)/SD. The Z score was used to evaluate the enrichment of a certain ORF in the CD44-high population, compared with the unsorted population. A higher Z score indicated an enhanced capability for an ORF to promote the conversion of HMLER cells to the CD44-high state.

## RNA-sequencing library preparation and data analysis

To prepare libraries for RNA sequencing of HME cells that overexpress HcRed, EGFP, QKI, RBFOX1 or SNAI1, we first extracted total RNA using the RNeasy Mini Kit (QIAGEN). Next, 1.5 ug of total RNA was used to generate first strand cDNA using Oligo(dT)12–19 primers (Invitrogen) and Affinity-Script Multi-Temp Reverse Transcriptase (Agilent). Second strand cDNA was synthesized using the NEBNext mRNA Second Strand Synthesis Module (NEB) and washed with AMPure XP beads (Beckman Coulter). Finally, libraries were generated from cDNA using the Nextera XT DNA Sample Preparation Kit (Illumina) and Nextera XT Indexes (Illumina). Libraries were pooled and sequenced on the

Illumina NextSeq 500 sequencer (paired-end, 150 bp). Image analysis and base calling were done using the standard Illumina pipeline, and then demultiplexed into FASTQ files. Reads were first trimmed using Trimmomatic (version 0.33) to remove Nextera adapter sequences down to a uniform length of 100nt (for compatibility with downstream splicing analysis software). Trimmed reads were then aligned to the human genome (hg19/GRCh37) using STAR (version 2.5.2b) (*Dobin et al., 2013*) and Gencode V19 gene annotations. Alternative splicing was quantified using rMATS (version 3.2.5) (*Shen et al., 2014*) by comparing each ORF to EGFP, with at least two to three replicates per group. The output from rMATS was further filtered to include only events for which the sum of inclusion counts (IC) and skipping counts (SC) was greater or equal to 10 for both sets of samples.

Alternative splicing quantification across cell lines in CCLE and TCGA breast invasive carcinoma was performed using JuncBASE v.0.8 with default parameters after initial sequence alignment using TopHat v1. To incorporate potentially novel exons, Cufflinks de novo transcript annotations were included from the CCLE data only.

## RNA preparation and polymerase chain reaction analysis

Total RNA was isolated using the RNeasy Mini kit (Qiagen, 74104) according to the manufacturer's protocol. A cDNA sample, prepared from 1 μg total RNA, was used for quantitative reverse transcription polymerase chain reaction (RT-PCR) performed with the High Capacity cDNA Reverse Transcription Kit (Life Technologies, 4368814) or iScript Reverse Transcription Supermix (BIO-RAD, 1708840). Quantitative PCR (qPCR) was done with the Power SYBR Green Master Mix (Life Technologies; 4368708); data were collected and analyzed on a Bio-Rad Real-Time PCR Detection System or a Roche LightCycler 480 qPCR instrument. Thermal-cycling parameters for the PCR were as follows: 95°C for 10 min, followed by 45 cycles each of 95°C for 20 s, 60°C for 60 s. The relative quantity of mRNA was normalized against the relative quantity of *RPLP0* or *GAPDH* mRNA in the same sample. Primer sequences in a 5′ to 3′ orientation are shown in *Supplementary file 1*.

## EMT signature analysis for CCLE cell lines and TCGA breast cancer samples

EMT UP and DOWN signatures were derived from previously published datasets based on their pattern of expression relative to the EMT phenotype (TAUBE_EMT_UP/DN, EMT gene set (*Taube et al., 2010*), GROGER_EMT_UP/DN, EMT gene set (*Gröger et al., 2012*), BYERS_EMT_UP/ DN (*Byers et al., 2013*). The EMT signature scores across CCLE were generated by using the ssGSEA algorithm (*Subramanian et al., 2005*). These scores were used to identify the top associated splice targets based on degree of association, an information-theoretic measure Information Coefficient (IC) (*Kim et al., 2016*). An empirical permutation test was performed for statistical significance calculations.

RNA-sequencing data of TCGA Breast invasive carcinoma (BRCA) samples were downloaded from the GDAC portal of the Broad Institute (http://gdac.broadinstitute.org/). The EMT signature scores across TCGA_BRCA samples were generated by the ssGSEA algorithm based on a previously published EMT gene expression signature (CHARAFE_EMT_UP and _DOWN combined) (*Charafe-Jauffret et al., 2006*; *Subramanian et al., 2005*). The top 20% of samples (total n = 1212, mesenchymal tumor = 242) that had the highest EMT scores were counted as mesenchymal tumor samples and the top 20% of samples (n = 242) that had the lowest EMT scores were counted as epithelial tumor samples. The gene expression of EMT markers and RBPs were compared between these mesenchymal and epithelial samples in *Figure 3G*. In addition, these scores were used to identify the top correlated gene expression based on degree of association by calculating Pearson Correlation Coefficiency (PCC) and their p values. Breast cancer subtypes were obtained from a PAM50 gene signature-based TCGA analysis (*Ciriello et al., 2015*) and correlated with the expression of specific isoforms.

## Gene expression analysis for murine mammary tumor gene expression

RNA sequencing data for gene expression in primary and recurrent MMTV-HER2 mammary tumors were previously published (*Goel et al., 2016*). In this model, withdrawal of HER2 expression leads to primary mammary tumor regression but is eventually followed by recurrence of HER2-resistant tumors that harbor a mesenchymal phenotype (*Figure 2—figure supplement 1D–F*). Ten out of 11

such recurrent tumors underwent EMT as shown by the expression of mesenchymal markers and the spindle-like cellular morphology (*Figure 2—figure supplement 1D*) (*Goel et al., 2016*). Strikingly, based on an analysis of the RNA-sequencing results from *Goel et al. (2016)*, we found that the expression of Qk (mouse homolog of human QKI) and Rbfox1 were significantly upregulated in the recurring mesenchymal mammary tumors relative to their expression in the corresponding, initially formed epithelial tumors (*Figure 2—figure supplement 1D,E*). The differential gene expression was evaluated by p values calculated by student's t-test of the normalized expression values between the recurrent tumors and primary tumors. The false discovery rate (FDR) values were generated by comparative marker selection analysis in Genepattern (*Reich et al., 2006*).

## Protein extraction and immunoblotting

Cell extract preparation and immunoblotting were completed as described (*Li et al., 2013*). All antibodies used for immunoblotting were listed in *Supplementary file 1*.

For RBFOX1 immunoblotting, we detected a 42 kDa band in HME cell lysate (predicted size). RBFOX1 levels are higher in HMLER cells and we observed a 33 kDa lower band, in addition to the 42 kDa band that corresponds to the predicted size of RBFOX1 (*Figure 2—figure supplement 2C, D*), which is likely a cleaved form of RBFOX1.

## Immunoprecipitation

For preparation of whole cell extract, HME or 293 T cells were harvested, and lysed on ice for 30 min with IP buffer containing 50 mM Tris HCl pH 7.0, 150 mM NaCl, 1 mM EDTA/pH 8, 0.5% Na-deoxycholate, 0.5% NP-40 and 10% glycerol, with protease inhibitors added before use. The lysates were sonicated with 10 pulses on ice, then centrifuged at 14,000 rpm for 5 min, and the supernatants were collected for immunoprecipitation. For preparation of nuclear protein, HMLE nuclear extract was prepared using the Nuclear Complex Co-IP kit (Active Motif #54001) as described in manufacturer's instructions. Briefly, cell pellets were resuspended in hypotonic buffer to break cell membrane and the nucleus were isolated by centrifugation. The nuclear fraction was further lysed by digestion buffer with enzymatic shearing cocktail before it was further diluted in the IP buffer provided in the kit.

For immunoprecipitations, QKI antibody (Bethyl Laboratories # A310-050A) or FLNB antibody (Millipore # AB9276) was added at a concentration of 1 ug per 1 mg of cell lysate, and the lysates were incubated for 2 hr at 4°C; protein A/G agarose was then added and lysates were further incubated for 2 hr at 4°C on a rotator. Protein A/G beads were washed four times in cold IP buffer followed by centrifugation. Samples were boiled in SDS loading buffer, and separated on an SDS-PAGE gel, followed by immunoblotting. For QKI immunoprecipitations, to digest the RBP-associated RNAs, cell lysate was incubated with 50 ng/ml of RNase at room temperature for 20 min before antibodies were added.

## siRNA transfection

HME, HMLE or HMLER cells were transfected with siRNAs, using Lipofectamine RNAi-MAX. Six hours before the siRNA transfection, cells were split into six-well plates. To prepare transfection complexes, 5 ul of RNAiMAX was mixed into 150 ul of OptiMEM medium in one tube, while 5 ul of 20 mM siRNA was mixed into 150 ul of OptiMEM medium in another tube. Tubes were incubated at room temperature for 20 min before being added to the cells. The cells were harvested 72 hr after transfection.

## Immunofluorescence microscopy

Immunofluorescence procedures have been described previously (*Li et al., 2013*). Briefly, HMLE cells were fixed with cold methanol for 2 min and permeabilized with PBS-1% Triton X-100 for 5 min. Cells were blocked in PBS-donkey serum for 1 hr before being incubated with primary antibody for 2 hr. Alexafluor 488-conjugated donkey anti-rabbit IgG (Invitrogen # R37118) was used as a secondary antibody. DNA was stained with DAPI. Images were acquired with a Nikon inverted microscope. For Phalloidin staining, cells were fixed with Formalin for 8 min and incubated with Phalloidin-Alexafluor 488 for 1 hr at room temperature before DAPI staining and image analysis.

## eCLIP-sequencing library preparation and data analysis

RBP-RNA interactions were crosslinked by UV exposure (254 nm, 400 mJ/cm$^2$) using a Spectrolinker XL-1500 UV crosslinker. eCLIP was then performed as previously described (*Van Nostrand et al., 2016*) (ENCODE protocol v1.P 20151108) with some minor modifications as follows: (1) immunoprecipitated RNA was 3' end ligated to a custom RNA adapter ('3 SR_RNA'); (2) RNA was released from the nitrocellulose membrane after transfer by treatment with 200 ul of an SDS solution (100 mM Tris, pH 7.5; 50 mM NaCl; 1 mM EDTA; 0.2% SDS) containing 10 ul of proteinase K (Life Technologies, AM2546) and incubating in an Eppendorf thermomixer (60 min at 50°C: 15 s at 1000 r.p.m., 30 s rest), as described in the irCLIP protocol (*Zarnegar et al., 2016*); (3) reverse transcription was done with a custom RT primer ('SR_RT'); (4) the 3' end of the cDNA was ligated to a custom DNA adapter ('SR_DNA'); (4) amplification of ligated cDNA was done with NEBNext Multiplex Oligos for Illumina (NEB, E7335S); (5) PCR amplified libraries were purified twice with AMPure XP beads (1.0X both times) and then directly quantified by qPCR and run on a Bioanalyzer High Sensitivity DNA chip, before being pooled and submitted for sequencing on the Illumina NextSeq 500 (single-end, 75 bp). For each RBP (QKI and RBFOX1), we prepared and sequenced two replicates and a single size-matched input control derived from the first replicate.

Sequenced reads were processed as previously described (*Van Nostrand et al., 2016*) (ENCODE pipeline v1.P 20160215), with some minor modifications. First, the unique molecular index (UMI) from the 5' end of each read was extracted using UMI Tools (parameters: umi_tools extract –bc-pattern=NNNNN). Next, adapters were trimmed using cutadapt (parameters: cutadapt –match-read-wildcards –times 1 -e 0.1 -O 1 –quality-cutoff 6 m 18 -a NNNNAGATCGGAAGAGCACACGTC TGAACTCCAGTCAC). Trimmed reads were first aligned to a database of human repetitive elements (RepBase) using STAR (v2.5.2b) (parameters: STAR –runMode alignReads –genomeDir/path/to/ RepBase –readFilesCommand zcat –outSAMunmapped Within –outFilterMultimapNmax 30 –outFilterMultimapScoreRange 1 –outSAMattributes All –outStd BAM_Unsorted –outSAMtype BAM Unsorted –outFilterType BySJout –outReadsUnmapped Fastx –outFilterScoreMin 10 –outSAMattrRGline ID:foo –alignEndsType EndToEnd). Reads not mapping to RepBase were then aligned to the human reference genome (hg19 with Gencode V19 annotations) using STAR (v2.5.2b) (parameters: STAR –runMode alignReads –genomeDir/path/to/hg19 –readFilesIn –outSAMunmapped Within –outFilterMultimapNmax 1 –outFilterMultimapScoreRange 1 –outStd BAM_Unsorted –outSAMattributes All –outSAMtype BAM Unsorted –outFilterType BySJout –outReadsUnmapped Fastx –outFilterScoreMin 10 –outSAMattrRGline ID:foo –alignEndsType EndToEnd). PCR duplicates were then removed using UMI Tools (parameters: umi_tools dedup –method directional-adjacency –spliced-is-unique) leaving only uniquely mapping reads. Finally, CLIP peaks were called using CLIPper software (*Lovci et al., 2013*) and identified peaks were normalized to the appropriate size-matched input control with 'Peak_input_normalization_wrapper.pl' (https://github.com/YeoLab/gscripts) before being merged between replicates. Binding motifs were identified using MEME-ChIP (*Machanick and Bailey, 2011*).

## Statistics and reproducibility

All data represent the average of at least three independent experiments, unless otherwise indicated. Significance was calculated by two-tail Student's t-test, using GraphPad software. Differences were considered significant when p was <0.05.

## Primer and oligo sequences

All primer and oligo sequences are listed in *Supplementary file 1*.

## Data availability

Both the RNA-seq data and the CLIP-seq data are deposited at NCBI Gene Expression Omnibus (accession number GSE98210).

# Acknowledgements

We thank Andy Aguirre, Joyce O'Connell, Nicole Burr, Hans Widlund, Bärbel Schröfelbauerand other members of the Hahn lab and Weinberg lab for helpful discussion. We thank Dr. Arnoud

Sonnenberg (Netherlands Cancer Institute) for providing expression plasmids for two isoforms of FLNB. This work was supported in part by US NIH grants R01 CA130988, U01 CA176058, a Susan G Komen Breast Cancer Postdoctoral Fellowship - Basic and Translational (PDF14300517) (JL), a Terri Brodeur Breast Cancer Foundation Grant (JL) and an NIH Pathway to Independence Award (K99 CA208028) (PSC). WCH is a consultant for Novartis.

## Additional information

### Funding

| Funder | Grant reference number | Author |
| --- | --- | --- |
| Susan G. Komen | PDF14300517 | Ji Li |
| Terri Brodeur Breast Cancer Foundation | | Ji Li |
| National Cancer Institute | K99 CA208028 | Peter S Choi |
| National Cancer Institute | R01 CA130988 | William C Hahn |
| National Cancer Institute | U01 CA176058 | William C Hahn |

The funders had no role in study design, data collection and interpretation, or the decision to submit the work for publication.

### Author contributions

Ji Li, Conceptualization, Resources, Data curation, Formal analysis, Funding acquisition, Validation, Investigation, Visualization, Methodology, Writing—original draft, Project administration, Writing—review and editing; Peter S Choi, Data curation, Formal analysis, Funding acquisition, Validation, Investigation, Visualization, Methodology, Writing—review and editing; Christine L Chaffer, Conceptualization, Investigation, Writing—review and editing; Katherine Labella, Justin H Hwang, Nina Ilic, Seav Huong Ly, Chao Dai, Kimberly Hagel, Andrew L Hong, Ole Gjoerup, Investigation; Andrew O Giacomelli, Jong Wook Kim, Angela N Brooks, Data curation, Formal analysis; John G Doench, Data curation, Investigation; Shom Goel, Jean J Zhao, Resources; Jennifer Y Ge, Formal analysis; David E Root, Supervision; Robert A Weinberg, Supervision, Investigation, Writing—review and editing; William C Hahn, Conceptualization, Resources, Data curation, Formal analysis, Supervision, Funding acquisition, Investigation, Methodology, Writing—original draft, Project administration, Writing—review and editing

### Author ORCIDs

Ji Li http://orcid.org/0000-0002-9075-0432
Peter S Choi http://orcid.org/0000-0002-2820-3032
Andrew O Giacomelli http://orcid.org/0000-0003-2109-0458
Andrew L Hong http://orcid.org/0000-0003-0374-1667
William C Hahn http://orcid.org/0000-0003-2840-9791

### Ethics

Animal experimentation: This study with use of animals was performed in accordance to the protocol (04-101) approved by Dana-Farber Cancer Institute (DFCI)'s Institutional Animal Care and Use Committee (IACUC). The animals were handled according to the Guide for the Care and Use of Laboratory Animals of the National Institute of Health.

### Decision letter and Author response

Decision letter https://doi.org/10.7554/eLife.37184.033
Author response https://doi.org/10.7554/eLife.37184.034

## Additional files

### Supplementary files

• Supplementary file 1. Antibodies, primer and oligo sequence This file contains information for antibodies, and the sequence for primers and oligonucleotides that were used in this study.
DOI: https://doi.org/10.7554/eLife.37184.026

• Transparent reporting form
DOI: https://doi.org/10.7554/eLife.37184.027

### Data availability

Both the RNA-seq data and the CLIP-seq data are deposited at NCBI Gene Expression Omnibus (accession number GSE98210).

The following dataset was generated:

| Author(s) | Year | Dataset title | Dataset URL | Database, license, and accessibility information |
|-----------|------|---------------|-------------|--------------------------------------------------|
| Choi P | 2017 | Alternative splicing regulated by QKI and RBFOX1 promotes the mesenchymal cell state in breast cancer | http://www.ncbi.nlm.nih.gov/geo/query/acc.cgi?acc=GSE98210 | Publicly available at the NCBI Gene Expression Omnibus (accession no. GSE98210) |

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
