## [Decision Letter]

Thank you for submitting your article "An alternative splicing switch in FLNB promotes the mesenchymal cell state in breast cancer" for consideration by *eLife*. Your article has been reviewed by three peer reviewers, and the evaluation has been overseen by Douglas Black as the Reviewing Editor and James Manley as the Senior Editor. The reviewers have opted to remain anonymous.

The reviewers have discussed the reviews with one another and the Reviewing Editor has drafted this decision to help you prepare a revised submission.

Summary:

This study by Li and colleagues examines the role of splicing regulation in driving the epithelial mesenchymal transition (EMT). EMT is a key event in the progression of tumors to a metastatic state and is controlled by an array of genetic mechanisms, including the transcription factors SNAI1, Twist, ZEB1 and others, as well as splicing regulators such as ESRP, RBFOX2 and QKI. The authors previously developed a breast cancer model by transforming mammary epithelial cells with oncogenic factors. The derived cells (HMLER) can be segregated into pre and post EMT populations based on CD44 expression, a marker of metastasis. The CD44 high cells express other mesenchymal markers and exhibit additional properties of these cells, including mammosphere formation. To identify drivers of the transition, the authors infected CD44 low cells with a bar coded cDNA expression library and cells were sorted for CD44 expression. Barcodes enriched in the CD44 high population identified candidates for cDNAs that can induce EMT in HMLER cells. These included the known EMT factor, SNAI1, an indication that other hits in the screen could be similarly important for EMT. These other hits included a preponderance of RNA binding proteins (RBP), two of which, RBFOX1 and QKI, can induce EMT. The authors show that RBFOX1 and QKI alone can increase mesenchymal marker expression and stimulate mammosphere formation. These RBPs are upregulated by SNAI1, and depletion of these proteins by RNAi or CRISPR reduces the effect of SNAI1 on CD44 expression and mammosphere formation. QKI and to a lesser extent RBFOX1 are upregulated in other breast tumors and tumor progression models. Using RNAseq the authors identify changes in splicing induced by QKI and RBFOX1 in their cells and find exons that are coregulated by these RBPs, whose splicing changes are induced by SNAI1. Using CLIP to identify binding sites for QKI and RBFOX1, they show that the proteins bind to sequences adjacent some of the EMT regulated exons making them likely direct targets of QKI and RBFOX1. Focusing on an alternative exon in the actin binding protein Filamin B (FLNB), they show that its skipping correlates with the basal B type of breast cancer. The exon skipped isoform is more active in inducing mesenchymal markers. Using genome editing to force skipping of this exon, they observe modest increases in these markers, in the percent of CD44 high cells, and in mammosphere formation. FLNB was previously shown to bind to the transcription factor FOXC1. Through knockdown experiments the authors show that FOXC1 is needed for the marker expression induced by the RBPs and FLNB.

The reviewers all agreed that his study is a significant contribution to the growing literature on the roles of RBP's and posttranscriptional regulation in EMT and tumor progression. Some of the effects are not large, much smaller than SNAI1, leading to questions as to how central these new players are to EMT. Nevertheless, there is a lot of interesting data here, and the experiments are generally well performed and comprehensive. This work will be of broad interest to groups working on both RNA biology and cancer. However, there are a number of issues that need to be addressed before the paper can be considered for publication.

Essential revisions:

1) There is no characterization of which QKI and RBFOX1 isoforms came out of the screen. Both of these genes produce multiple isoforms, nuclear and cytoplasmic, that have different functions and can cross regulate each other. One cannot assess the true targets of whatever forms were isolated from the library without knowing their structure. Similarly, there needs to be an assessment of whether the endogenous Rbfox or QKI isoforms change with EMT or with ectopic expression of a particular isoform. It appears that the RBFOX1 isoforms could change between HME cells and HMLER cells from the immunoblots in Figure 2—figure supplement 2.

2) In Figure 2, did the cell proliferation change when the candidate RNA binding proteins were overexpressed? Were there any morphological differences between the CD44-high cells induced by SNAI1 overexpression and those induced by overexpression of QKI or RBFOX1?

3) RBFOX2 is expressed in a wide variety of cell types and was previously implicated in EMT. In contrast, RBFOX1 is reported to be most abundant in heart, muscle and brain. It is thus important to show validation of the RBFOX1 antibodies, confirming that they do not cross-react with RBFOX2. The isolation of RBFOX1 from an ectopic expression screen is not so surprising, but why was RBFOX2 not isolated? Is it not active for inducing EMT? Given the previous data on RBFOX2 and EMT, it seems surprising that it does not have similar activity.

4) Many RNA binding proteins co-immunoprecipitate because they bind to common RNAs. The interaction between QKI and RBFOX1 described in Figure 4—figure supplement 1C is not meaningful unless the sample has been treated with RNase.

5) In Figure 6, the effects of the two FLNB isoforms are relatively small, especially in the CRISPR experiments. It is difficult to draw conclusions from these data, and some of these results appear to be contradictory. In Figure 6C, FLNB-ΔH1 is shown to induce EMT marker expression. But the model in Figure 7 seems to be that the ΔH1 isoform sequesters less of the FOXC1 transcription factor leading to more expression of the EMT markers. If this were true, then over expression of FLNB-L should reduce marker expression and FLNB-ΔH1 should have no effect. The reduction in nuclear FLNB presented in Figure 7—figure supplement 1A is difficult to see and needs to be quantified. The proposed model does not agree with the reported data.

6) In Figure 6, the protein levels of Filamin B should be shown after siRNA and CRISPR knockdown.

[Editors' note: further revisions were requested prior to acceptance, as described below.]

Thank you for resubmitting your work entitled "An alternative splicing switch in FLNB promotes the mesenchymal cell state in human breast cancer" for further consideration at *eLife*. Your revised article has been favorably evaluated by James Manley (Senior Editor) and Reviewing Editor Douglas Black.

In this revised manuscript from Li and colleagues, the authors have added extensive new data and addressed nearly all of the concerns raised by the reviewers. The manuscript is largely ready for publication in *eLife*. However, the editors request that one final point be addressed. The RBFOX1 isoform isolated from the screen and used in the analysis, NM_145893.2, appears to contain the third to last exon (hg38 chr16:7,693,315-7,693,367). Since isoforms containing this exon have been found to be largely localized to the cytoplasm (PMID: 15824060; PMID: 19762510), this raises questions regarding the mechanisms leading to the RBFOX1 dependent splicing change. It is possible that the localization may be different in the system studied here, but this should be confirmed and described. Alternatively, if the RBFOX1 protein driving EMT is cytoplasmic as seen previously, the authors should note this and revise their model in the text.

---

## [Author Response]

Essential revisions:1) There is no characterization of which QKI and RBFOX1 isoforms came out of the screen. Both of these genes produce multiple isoforms, nuclear and cytoplasmic, that have different functions and can cross regulate each other. One cannot assess the true targets of whatever forms were isolated from the library without knowing their structure. Similarly, there needs to be an assessment of whether the endogenous Rbfox or QKI isoforms change with EMT or with ectopic expression of a particular isoform. It appears that the RBFOX1 isoforms could change between HME cells and HMLER cells from the immunoblots in Figure 2—figure supplement 2.

In the open reading frame (ORF) library (human ORFame v8.1, Broad Institute), the ORF encoding QKI is the transcript NM_006775.2 (transcript variant 1, also known as QKI-5). The ORF encoding RBFOX1 is NM_145893.2 (transcript variant 3, also known as RBFOX1 beta). We have included this information in the revised manuscript (subsection “Expression of QKI and RBFOX1 are necessary and sufficient to induce an intermediate mesenchymal cell state”, first paragraph).

We agree that some RNA binding proteins such as MBNL1 and RBFOX2 undergo alternative splicing during EMT (Shapiro et al., 2011; Venables et al., 2013). To address whether endogenous QKI undergoes alternative splicing during EMT, we queried the RNA sequencing data that we generated in HME cells in which we expressed SNAI1. We found no significant changes in the distribution of endogenous QKI isoforms (main Refseq transcripts) upon expression of either RBFOX1 or SNAI1 (Author response image 1).

For RBFOX1, we found that our RNA-sequencing data was of insufficient depth to adequately detect RBFOX1 isoform expression and therefore performed RT-PCR to assess alternative N-terminal exons and mutually exclusive exons to detect alternative splicing of internal cassette exons. We found that ectopic expression of QKI or SNAI1 did not result in any significant changes in the exons that we examined (Author response image 1).

**Author response image 1. respfig1:** Endogenous QKI and RBFOX1 isoform expression in HME and HMLER cells. (**A**) Expression of endogenous QKI isoforms based on RSEM quantification of RNA-sequencing data in HME cells expressing control EGFP, RBFOX1, or SNAI1. (**B**) RT-PCR for alternatively spliced exons in endogenous RBFOX1 in HME cells expressing control EGFP, SNAI1 or QKI. (**C**) Immunoblots for RBFOX1 protein expression in HME or HMLER cells.

As noted by the reviewers, we observed a strong smaller RBFOX1 protein band in immunoblotting the HMLER cell lysate compared with the HME cell lysate (previous Figure 2—figure supplement 2). To confirm this finding, we performed a side-by-side comparison of the RBFOX1 proteins in the HME and HMLER cell lysate (Author response image 1). As we found previously, RBFOX1 levels are higher in HMLER cells and showed a stronger 33 kDa lower band, in addition to the 42 kDa band that corresponds to the predicted size of RBFOX1. Through a genetic knockout experiment (see the response to major point #3), we confirmed that both bands are forms of RBFOX1 (Author response image 1 and Author response image 2). Since we failed to detect a shorter mRNA form of RBFOX1 (see above), it is likely that this short form of RBFOX1 was generated through post-translational modifications, possibly through endogenous proteolytic cleavage of RBFOX1 proteins. Using Expasy Peptidecutter tool (web.expasy.org/peptide_cutter/), we found that there are at least 20 proteolytic sites in RBFOX1 that may generate a shorter 33 kDa protein. Further experiments will be needed to determine the identity of the shorter RBFOX1 protein product detected in HMLER cells. We have included a description for RBFOX1 immunoblotting patterns in the revised manuscript (subsection “Protein extraction and immunoblotting”).

**Author response image 2. respfig2:** Verification of RBFOX1 antibody specificity. RBFOX1 (**A**) or RBFOX2 (**B**) was depleted by CRISPR-Cas9-mediated knock out in HMLE cells. RBFOX1 and RBFOX2 proteins were detected by the corresponding antibodies.

2) In Figure 2, did the cell proliferation change when the candidate RNA binding proteins were overexpressed? Were there any morphological differences between the CD44-high cells induced by SNAI1 overexpression and those induced by overexpression of QKI or RBFOX1?

In response to the reviewers’ question, we overexpressed EGFP, QKI, RBFOX1 and SNAI1 in HME cells and measured cell number after 7 days. We found that overexpression of QKI and RBFOX1 reduced cell proliferation by 40% and 45% respectively, while overexpression of EMT transcription factor SNAI1 decreased cell proliferation by 50%. These observations confirm prior work that showed that initiation of the EMT program slows cell proliferation (Tsai et al., 2012; Vega et al., 2004) and support that notion that expression of QKI and RBFOX1 induces the EMT program. We thank the reviewers for raising this important point and have now included these experiments in Figure 2—figure supplement 3C of the revised manuscript.

The reviewers also asked whether there are morphological differences between the CD44-high cells induced by SNAI1 overexpression and those induced by overexpression of QKI or RBFOX1. To address this question, we sorted for CD44-high HMLER cells induced by QKI, RBFOX1 and SNAI1 expression and stained for phalloidin (marker for F-actin filament) for better visualization of cell morphology. Compared with the control HMLER CD44-low cells, the CD44- high cells induced by QKI, RBFOX1 and SNAI1 all showed similar cell morphologies marked by elongated and spindle shaped cells (revised Figure 2—figure supplement 3A). Thus, despite the relatively lower occurrence of the CD44-high population in QKI- and RBFOX1- expressing cells comparing with SNAI1- expressing cells, the CD44-high cell populations generated by QKI, RBFOX1 or SNAI1 are morphologically similar. We now include these observations in the revised manuscript (Figure 2—figure supplement 3A).

3) RBFOX2 is expressed in a wide variety of cell types and was previously implicated in EMT. In contrast, RBFOX1 is reported to be most abundant in heart, muscle and brain. It is thus important to show validation of the RBFOX1 antibodies, confirming that they do not cross-react with RBFOX2. The isolation of RBFOX1 from an ectopic expression screen is not so surprising, but why was RBFOX2 not isolated? Is it not active for inducing EMT? Given the previous data on RBFOX2 and EMT, it seems surprising that it does not have similar activity.

The reviewers are correct that RBFOX2 has been reported to regulate EMT (Venables et al., 2013; Braeutigam et al., 2014). Unfortunately, the RBFOX2 clone (TRCN0000478244) included in the human ORFome 8.1 (Broad Institute) that we used for the screen harbors three mutations (a 396-449 deletion, a 752-763 deletion and a C to T substitution at 1007), which apparently disrupts the function of RBFOX2 and likely explains why we did not isolate RBFOX2. We have added this information to the revised manuscript (subsection “A genome scale ORF screen to identify regulators of the mesenchymal cell state”, last paragraph).

To validate the specificity of the RBFOX1 antibodies, we used CRISPR-Cas9 editing to generate RBFOX1 and RBFOX2 knock-out cells and used them to test the antibody specificity (Author response image 2). We found that RBFOX1 antibody (Sigma SAB2100002) recognized a strong band at 42 kDa and a weak band at 33 kDa (see response to major point #1). The RBFOX2 antibody (Bethyl A300-864A) detected two bands at 52 kDa and 45 KDa (likely to be two splicing isoforms), which were not detected after genetic ablation of the corresponding genes. Therefore, we conclude that the antibodies that we used are specific for RBFOX1.

4) Many RNA binding proteins co-immunoprecipitate because they bind to common RNAs. The interaction between QKI and RBFOX1 described in Figure 4—figure supplement 1C is not meaningful unless the sample has been treated with RNase.

We agree with the reviewers that this is an important specificity control. In response, we incubated the cell lysates with RNase at room temperature for 20 minutes to digest the RNAs before we performed the immunoprecipitation experiment using QKI antibody (revised Figure 4—figure supplement 1C). We found that RNase digestion did not substantially affect the association between QKI and RBFOX1. Therefore, we conclude that the interaction between QKI and RBFOX1 was not mediated through RNAs. We now include this figure in the revised manuscript.

*5) In Figure 6, the effects of the two FLNB isoforms are relatively small, especially in the CRISPR experiments. It is difficult to draw conclusions from these data, and some of these results appear to be contradictory. In Figure 6C, FLNB*-Δ*H1 is shown to induce EMT marker expression. But the model in Figure 7 seems to be that the ΔH1 isoform sequesters less of the FOXC1 transcription factor leading to more expression of the EMT markers. If this were true, then over expression of FLNB-L should reduce marker expression and FLNB-ΔH1 should have no effect. The reduction in nuclear FLNB presented in Figure 7—figure supplement 1A is difficult to see and needs to be quantified. The proposed model does not agree with the reported data.*

We thank the reviewers for raising this important question, which prompted us to further investigate the mechanism by which FLNB alternative splicing regulates EMT. First, to further investigate the function of FLNB-L in regulating EMT, we suppressed FLNB expression by siRNAs in HMLE cells, in which the majority of FLNB protein is the FLNB-L isoform. We found that suppression of FLNB-L upregulated the expression of the mesenchymal markers VIM and FN1, indicating that FLNB-L plays a negative role in regulating EMT (revised Figure 6C), consistent with the model in Figure 7.

Second, we investigated how FLNB-ΔH1 expression increased the expression of mesenchymal markers in the previous Figure 6D. We noted that in the previous Figure 6D, exogenous FLNB isoforms were expressed in the presence of the endogenous proteins. Importantly, it has been demonstrated that filamin proteins dimerize in cells (Berry et al., 2005; Pudas et al., 2005; Stossel et al., 2001). Therefore, it is possible that the exogenous FLNB-ΔH1 dimerized with the endogenous FLNB-L and affected the function of endogenous FLNB-L. Specifically, overexpression of FLNB-ΔH1 may promote the release of the nuclear FOXC1 by dimerizing with the endogenous FLNB-L. To test this possibility, we suppressed the endogenous FLNB using 2 siRNAs targeting the 3’UTR region of FLNB and rescued the FLNB expression by overexpression of two different FLNB isoforms (revised Figure 6D). Depletion of endogenous FLNB-L consistently promoted mesenchymal marker expression (Lane 2 from the left). In the context of endogenous FLNB suppression, FLNB-L expression reduced the expression of mesenchymal markers, FN1 and VIM (Lane 2, 3). In contrast, the expression of FLNB-ΔH1 did not decrease the mesenchymal marker expression (Lane 2-4). Taken together, these observations strongly suggest that FLNB-L plays a negative role in promoting EMT.

Third, as indicated by the reviewers and the new experiments that we described above, we expected to see a reduction of mesenchymal markers upon FLNB-L expression in mesenchymal cells. However, the cell line that we used in Figure 6C and D was human mammary epithelial cells (HMLE), which reside in an epithelial cell state. The effect of FLNB-L overexpression was not likely to make them even more epithelial. To further test this hypothesis, we overexpressed FLNB-L and FLNB-ΔH1 in a breast cancer cell line that exhibits the mesenchymal state, MDA-MD-231, and examined the expression of mesenchymal markers (new Figure 6—figure supplement 1A). We found that overexpression of FLNB-L in this mesenchymal cell line indeed reduced the expression of mesenchymal markers. Again, the expression of FLNB-ΔH1 did not seem to significantly further increase VIM expression probably because MDA-MD-231 cells are already in a mesenchymal state.

We thank the reviewers for prompting us to extend our understanding of the mechanism by which FLNB alternative splicing regulates EMT. We now include the new experiments and discussion in the revised manuscript (Figure 6C, D and Figure 6—figure supplement 1A, subsection “FLNB isoform switching promotes a mesenchymal - like cell state”, second paragraph). We also include the quantification of FLNB nuclear intensity in the revised Figure 7—figure supplement 1A (right).

6) In Figure 6, the protein levels of Filamin B should be shown after siRNA and CRISPR knockdown.

We now provide the immunoblots for Filamin B in the revised Figure 6F (bottom) and revised Figure 6—figure supplement 1D where we assess Filamin B protein levels after CRISPR- or siRNA-mediated change of the endogenous ratios of the two FLNB isoforms (FLNB-L and FLNB-ΔH1). Specifically, we performed two experiments. In the first experiment, we used CRISPR-mediated editing of the splicing junction of the genomic locus and found a minor reduction of the endogenous FLNB proteins. We also performed a second experiment in which we used siRNA-mediated suppression of a specific mRNA isoforms and found a significant reduction of the total FLNB proteins, indicating that the siRNA approach has a more profound impact on the total protein levels. In both methods, the ratio of the endogenous FLNB isoforms changed substantially.

[Editors' note: further revisions were requested prior to acceptance, as described below.]

In this revised manuscript from Li and colleagues, the authors have added extensive new data and addressed nearly all of the concerns raised by the reviewers. The manuscript is largely ready for publication in eLife. However, the editors request that one final point be addressed. The RBFOX1 isoform isolated from the screen and used in the analysis, NM_145893.2, appears to contain the third to last exon (hg38 chr16:7,693,315-7,693,367). Since isoforms containing this exon have been found to be largely localized to the cytoplasm (PMID: 15824060; PMID: 19762510), this raises questions regarding the mechanisms leading to the RBFOX1 dependent splicing change. It is possible that the localization may be different in the system studied here, but this should be confirmed and described. Alternatively, if the RBFOX1 protein driving EMT is cytoplasmic as seen previously, the authors should note this and revise their model in the text.

To determine the subcellular localization of this RBFOX1 isoform (NM_145893.2), we performed immunofluorescence for the RBFOX1 ORF that we used in these experiments (v5 tagged). In consonance with the prior report in neuronal cells [Figure 2, (Lee et al., 2009)], we found that the 62% of this protein isoform localized to the cytoplasm in human mammary epithelial cells. Meanwhile, 38% of RBFOX1 localizes to the nucleus (revised Figure 7—figure supplement 1D). This observation confirms that a substantial fraction of this RBFOX1 isoform is in the nucleus. In addition, our eCLIP data (Figure 4 in the manuscript) also suggested that a portion of the RBFOX1 isoform studied here binds to specific mRNAs. Therefore, although it remains possible that the cytoplasmic fraction of RBFOX1 may also contribute to EMT, the nuclear fraction of this RBFOX1 isoform likely plays an important role in regulating EMT-related pre-mRNA splicing. We have included a paragraph in the revised manuscript (Discussion, fourth paragraph; Figure 7—figure supplement 1D) to discuss this observation. We thank the editor for raising this important point.